# An anisotropic van der Waals dielectric for symmetry engineering in functionalized heterointerfaces

Zeya Li [1,2,11], Junwei Huang [1,2,11], Ling Zhou [1,2,11], Zian Xu [3,11], Feng Qin [1,2], Peng Chen[1,2], Xiaojun Sun[1,2], Gan Liu[1,4], Chengqi Sui[1,2], Caiyu Qiu[1,2], Yangfan Lu [5], Huiyang Gou [6], Xiaoxiang Xi [1,4], Toshiya Ideue [7,8] ✉, Peizhe Tang [3,9] ✉, Yoshihiro Iwasa [7,10] & Hongtao Yuan [1,2] ✉

Van der Waals dielectrics are fundamental materials for condensed matter physics and advanced electronic applications. Most dielectrics host isotropic structures in crystalline or amorphous forms, and only a few studies have considered the role of anisotropic crystal symmetry in dielectrics as a delicate way to tune electronic properties of channel materials. Here, we demonstrate a layered anisotropic dielectric, $SiP_2$, with non-symmorphic twofold-rotational $C_2$ symmetry as a gate medium which can break the original threefold-rotational $C_3$ symmetry of $MoS_2$ to achieve unexpected linearly-polarized photoluminescence and anisotropic second harmonic generation at $SiP_2/MoS_2$ interfaces. In contrast to the isotropic behavior of pristine $MoS_2$, a large conductance anisotropy with an anisotropy index up to 1000 can be achieved and modulated in $SiP_2$-gated $MoS_2$ transistors. Theoretical calculations reveal that the anisotropic moiré potential at such interfaces is responsible for the giant anisotropic conductance and optical response. Our results provide a strategy for generating exotic functionalities at dielectric/semiconductor interfaces via symmetry engineering.

Symmetry breaking in low dimensional heterostructures can provide unprecedented possibilities to generate emergent quantum phenomena in condensed matter physics[1–8]. In general, van der Waals (vdW) dielectric at atomically-sharp semiconductor/dielectric interfaces can break the symmetry of the target materials and form moiré patterns with specific lattice mismatch[2,9,10], exhibiting remarkable capabilities to control electronic states and further realize exotic quantum

phenomena therein. Examples of these interfacial phenomena such as Chern insulating states[11,12], charge density wave states[13,14] and topological valley currents[15] have been demonstrated in the heterointerfaces based on $h$-BN dielectric[2,9,11–17], in which both $h$-BN dielectric and channel materials (graphene or transition metal dichalcogenides, TMDCs) show threefold-rotational symmetry ($C_3$) along the out-of-plane axis at their interface. In contrast, a low-symmetric dielectric

[1]National Laboratory of Solid State Microstructures, and Collaborative Innovation Center of Advanced Microstructures, Nanjing University, Nanjing 210093, China. [2]College of Engineering and Applied Sciences, and Jiangsu Key Laboratory of Artificial Functional Materials, Nanjing University, Nanjing 210023, China. [3]School of Materials Science and Engineering, Beihang University, Beijing 100191, China. [4]School of Physics, Nanjing University, Nanjing 210093, China. [5]College of Materials Sciences and Engineering, National Engineering Research Center for Magnesium Alloys, Chongqing University, Chongqing 400030, China. [6]Center for High Pressure Science and Technology Advanced Research, Beijing 100094, China. [7]Quantum Phase Electronic Center and Department of Applied Physics, The University of Tokyo, Tokyo 113-8656, Japan. [8]Institute for Solid State Physics, The University of Tokyo, Chiba 277-8581, Japan. [9]Max Planck Institute for the Structure and Dynamics of Matter, Center for Free Electron Laser Science, Hamburg 22761, Germany. [10]RIKEN Center for Emergent Matter Science, Hirosawa 2-1, Wako 351-0198, Japan. [11]These authors contributed equally: Zeya Li, Junwei Huang, Ling Zhou, Zian Xu. ✉e-mail: ideue@issp.u-tokyo.ac.jp; peizhet@buaa.edu.cn; htyuan@nju.edu.cn

material without $C_3$ symmetry (for example, with $C_2$ symmetry) can in principle break the $C_3$ symmetry in monolayer semiconductors by forming anisotropic moiré potentials at the interface[5] and result in exotic optical response and anisotropic electronic transport, while retaining the gating capability as a dielectric medium. Therefore, the vdW dielectrics with lower lattice symmetry can generate unique moiré physics and additional device functionalities at the symmetry-mismatched interfaces. However, an experimental confirmation of such a strategy remains elusive.

Herein, we demonstrate a unique anisotropic layered dielectric material $SiP_2$ and reveal its capability to generate giant anisotropy in optical response and electronic transport in isotropic TMDC semiconductors via symmetry engineering. We realize a high-performance $SiP_2$-gating $MoS_2$ transistor with large on/off ratios $>10^5$ and low leakage currents (far below the low power limit) and further observe an insulator-to-metal transition in $SiP_2$-gated 1L-$MoS_2$, indicating a great dielectric capability of $SiP_2$ material. Surprisingly, a linearly-polarized photoluminescence and an anisotropic second harmonic generation signals are observed in 1L-$MoS_2$/$SiP_2$ heterostructure, which are in sharp contrast to the isotropic features of pristine 1L-$MoS_2$. Remarkably, we find a large anisotropic conductance in the 1L-$MoS_2$/$SiP_2$ heterostructure and the tunable anisotropy index reaches a considerable value of 1000 with $SiP_2$ gating, which is among the largest

values reported so far, including intrinsically anisotropic materials[18]. Our first-principles calculations reveal that such giant anisotropy in optical response and electronic transport result from the generated anisotropic moiré potential in 1L-$MoS_2$/$SiP_2$ heterostructure that strongly renormalizes the structural and electronic properties of 1L-$MoS_2$ at the heterointerface. Note that the interfacial symmetry engineering by breaking the $C_3$ symmetry of 1L-$MoS_2$ using $SiP_2$ dielectric with $C_2$ symmetry can be regarded as a strategy to tune electronic properties of channel semiconductors and realize moiré physics at the heterointerfaces.

## Results

### $MoS_2$ transistors gated with $SiP_2$ dielectric

To evaluate the performance of the $SiP_2$ dielectric, we measured the transfer characteristics of $MoS_2$ transistors with dual-gate geometry in which 20-nm-thick $SiP_2$ and 300-nm-thick $SiO_2$ are used as top and bottom gate media (Fig. 1a, b and Supplementary Fig. 1). As shown in Fig. 1c, when sweeping the top gate voltage $V_{tg-SiP_2}$ up to 5 V, the 5-nm-thick $MoS_2$ transistor shows an on/off ratio as high as $10^5$, which is comparable to those values in $h$-BN-gated $MoS_2$ transistors (Supplementary Table 2) and meets the well-known criterion for practical logic circuit applications[19]. In contrast, when sweeping the bottom gate voltage $V_{bg-SiO_2}$ to -5 V, the transistor generates an on/off ratio as small

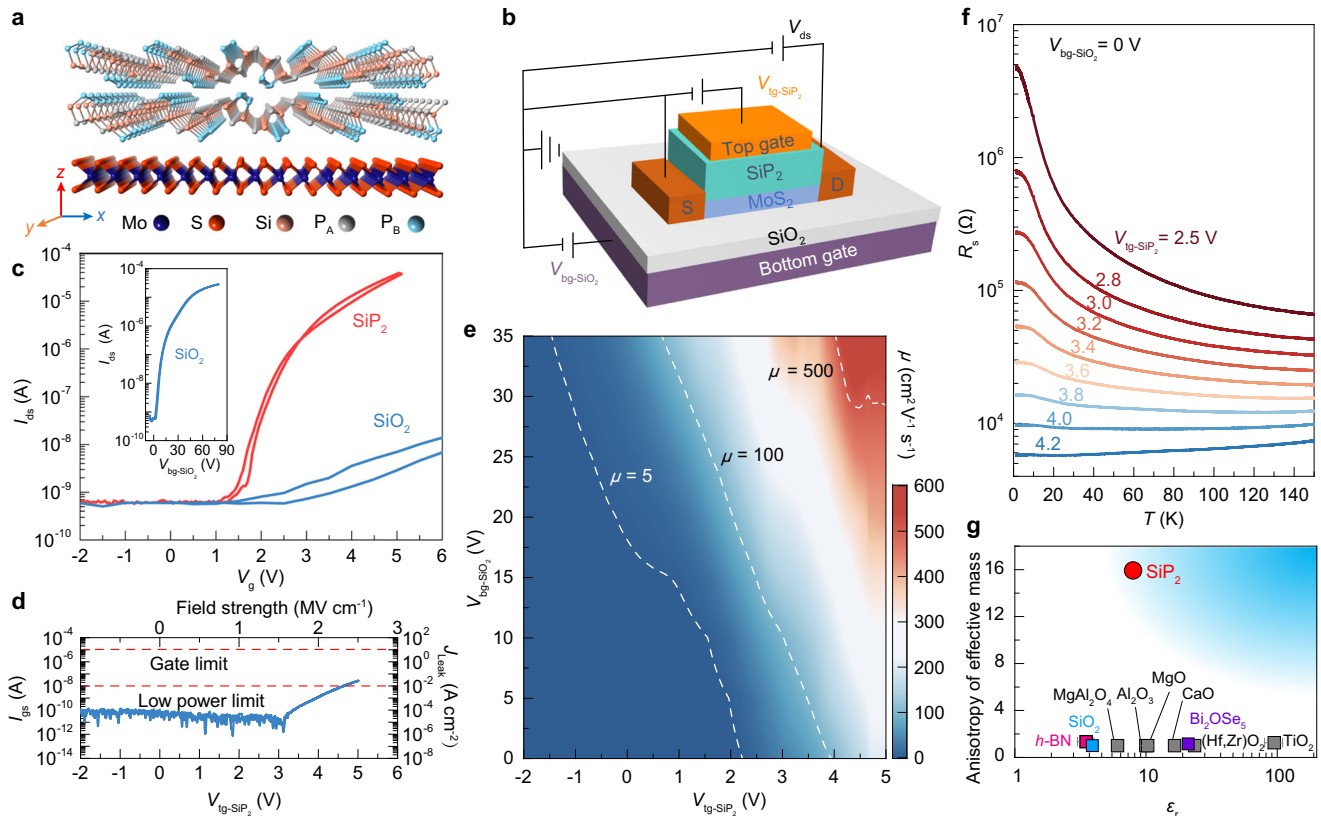

**Fig. 1 | High-performance $MoS_2$ transistors gated with a $SiP_2$ dielectric with non-symmorphic $C_2$ rotational symmetry. a** Schematic structure of the cross-section of a $MoS_2$/$SiP_2$ heterostructure. Orange, gray, light blue, red, and dark blue spheres represent Si, $P_A$, $P_B$, S, and Mo atoms, respectively. The $P_B$ atoms form the unique quasi-1D $P_B$–$P_B$ chains along the $y$ direction in $SiP_2$ crystal lattice. **b** Schematic illustration of $MoS_2$/$SiP_2$ dual-gated device. The top gate medium is 20-nm-thick $SiP_2$, and the bottom gate medium is 300-nm-thick $SiO_2$. D and S represent the drain and source electrodes. $V_{tg-SiP_2}$ and $V_{bg-SiO_2}$ are the top gate voltage and back gate voltage, respectively. And $V_{ds}$ is the source-drain voltage applied on the $MoS_2$ channel material. **c** Transfer curves of the 5-nm-$MoS_2$-based transistor at 2 K when sweeping top (red curve) and bottom (blue curve) gate voltages via the $SiP_2$ and $SiO_2$ dielectric media. The inset is a transfer curve with scanning $V_{bg-SiO_2}$.

**d** Leakage current $I_{gs}$ as a function of $V_{tg-SiP_2}$ on a 5-nm $MoS_2$ device at 2 K. $I_{gs}$ and $V_{tg-SiP_2}$ are rescaled to the leakage current density ($J_{Leak}$) and the electric field strength for a better comparison (right and top axes). Horizontal red lines mark the limits of leakage current density for various types of integrated circuits. **e** The field-effect mobility $\mu$ as a function of $V_{tg-SiP_2}$ and $V_{bg-SiO_2}$ for the 5-nm-$MoS_2$/$SiP_2$ device. The white dashed lines highlight that $\mu$ equals 5, 100, and 500 $cm^2$ $V^{-1}$ $s^{-1}$. **f** Sheet resistances ($R_s$) versus temperature of a 1L-$MoS_2$ device under different $V_{tg-SiP_2}$ values. **g** Comparison of the anisotropy ratio of the effective mass for dielectrics with different relative dielectric constants ($\varepsilon_r$). The anisotropy ratio is defined as the ratio of the electron effective mass ($m_e$) along the $y$ and $x$ directions of the corresponding lattice.

as 10, and requires a large $V_{bg-SiO_2}$ over 80 V to achieve an on/off ratio of $10^5$ (inset of Fig. 1c). This comparison directly demonstrates that, compared to $SiO_2$, the $SiP_2$ gate dielectric with larger dielectric constant and smaller thickness can achieve great capacitive capability. The measured leakage current of $SiP_2$-gated $MoS_2$ transistor is as small as approximately $10^{-5}$ A cm$^{-2}$ at an external electric field strength of 1.5 MV cm$^{-1}$ (Fig. 1d). Such a low leakage current is comparable to those of transistors gated by high-$\kappa$ dielectrics[19–21] such as $Al_2O_3$, $HfO_2$ or $Bi_2SeO_5$, and better than the criteria of the low-power limit and the standard complementary metal–oxide–semiconductor gate limit[19]. With increasing $V_{tg-SiP_2}$, the field-effect mobility $\mu$ at 2 K can reach ~600 cm$^2$ V$^{-1}$ s$^{-1}$ when $V_{bg-SiO_2}$ is fixed at 35 V (Fig. 1e). Even for $SiP_2$-gated 1L-$MoS_2$ transistors with the same device geometry (Supplementary Fig. 2), the mobility of 330 cm$^2$ V$^{-1}$ s$^{-1}$ at 2 K is better than those reported values in $HfO_2$-gated 1L-$MoS_2$ devices[22] (174 cm$^2$ V$^{-1}$ s$^{-1}$ at 4 K), indicating that the vdW $SiP_2$ material with a large dielectric constant can effectively reduce the charge scattering and increase the mobility of $MoS_2$ transistors. Such excellent performance with high on/off ratio, low leakage current, and high mobility in $MoS_2$/$SiP_2$ devices suggests that layered $SiP_2$ can be a high-performance dielectric in switching devices.

To demonstrate the great gate tunability of $SiP_2$ dielectric, we measured the temperature-dependent sheet resistance ($R_s$–$T$) and observed a gating-induced insulator–metal transition in the 1L-$MoS_2$/$SiP_2$ transistor. As shown in Fig. 1f, the $R_s$–$T$ curves show typical insulating behavior with negative temperature coefficients $dR_s/dT$ and follow a thermal activation dependence[23] when $V_{tg-SiP_2} < 3.8$ V and $V_{bg-SiO_2} = 0$ V (Supplementary Fig. 3a). The extracted activation energy decreases monotonically from ~7 meV to near zero as $V_{tg-SiP_2}$ increases from 2.5 V to 3.8 V (Supplementary Fig. 3b). As a result, $R_s$ starts to decrease with cooling temperature and the positive $dR_s/dT$ shows the typical metallic behavior when $V_{tg-SiP_2} > 3.8$ V, directly indicating an insulator–metal transition[23] in $SiP_2$-gated 1L-$MoS_2$. Such a transition in $SiP_2$-gated 1L-$MoS_2$ transistor directly manifests the excellent dielectric property of layered $SiP_2$ as a gate medium to modulate the electronic states of ultrathin semiconductors.

To experimentally evaluate the dielectric constant of $SiP_2$, we measured the sheet carrier density ($n_{2D}$) of $MoS_2$ (5 nm) as a function of $V_{tg-SiP_2}$ based on Hall effect measurements. The $n_{2D}$ values of top-gated $MoS_2$ remain nearly unchanged below the threshold voltage of 1.7 V and can be continually modulated to $8 \times 10^{12}$ cm$^{-2}$ by increasing $V_{tg-SiP_2}$ to 5 V (Supplementary Fig. 4a). Note that the dual-gate-modulated $n_{2D}$ can reach a maximum value close to $10^{13}$ cm$^{-2}$ (Supplementary Fig. 4b). The relative dielectric constant $\varepsilon_r$ is evaluated to be 8.1 for $SiP_2$ by fitting the linear part of the $n_{2D}$–$V_{tg-SiP_2}$ data using $n_{2D} = \varepsilon_0 \varepsilon_r V_{tg-SiP_2}/(et_{SiP_2})$, where $e$ is the electron charge, $\varepsilon_0$ is the vacuum permittivity, and $t_{SiP_2} = 20$ nm is the thickness of $SiP_2$ (more details in "Methods"). Such a dielectric constant of 8.1 in layered $SiP_2$ is larger than those of $SiO_2$ and $h$-BN dielectrics[24,25] and comparable to that of $Al_2O_3$ dielectric[24] (Fig. 1g, a detailed comparison is given in Supplementary Tables 3 and 4), well consistent with the theoretical estimation from first-principles calculations[26].

As a typical vdW dielectric with excellent performance, another distinctive nature of $SiP_2$ is the anisotropic lattice structure with nonsymmorphic $C_2$ symmetry. In sharp contrast to the highly symmetric crystal structure of those widely used dielectrics (oxides and $h$-BN), such an anisotropic in-plane lattice structure of $SiP_2$ leads to a highly anisotropic ratio of electron effective masses (~16, Fig. 1g) for the band edge states in its electronic band structure[26], and provides an opportunity to engineer the interfacial symmetry of vdW heterostructure combined monolayer TMDCs with $SiP_2$. For example, the 1L-$MoS_2$/$SiP_2$ heterostructure shows no rotational symmetry and can exhibit in-plane anisotropic optical and electronic properties (details discussed below). In particular, if the zigzag direction of $MoS_2$ and the $P_B$–$P_B$ chain of $SiP_2$ (the direction parallel to the $P_B$–$P_B$ chain of

$SiP_2$ is defined as the $y$ direction, while the perpendicular direction is defined as the $x$ direction) are aligned in parallel (Fig. 2a, b), the mirror symmetry along the $x$ direction can remain in the 1L-$MoS_2$/$SiP_2$ heterostructure; otherwise, all crystal symmetries in $MoS_2$ are broken. The perturbation for the electronic structures of stacked heterostructures can be used to generate in-plane polarization and Berry curvature dipole at the interface[27], realizing emergent interfacial phenomena such as directional quantum shift current[5], nonlinear Hall effect[1] and circular photo-galvanic effect (Supplementary Fig. 5).

## $SiP_2$-induced anisotropic optical response in 1L-$MoS_2$

To understand the engineered symmetry of heterointerfaces, we performed second harmonic generation (SHG) measurements on 1L-$MoS_2$/$SiP_2$ heterostructures under a parallel measurement geometry (Fig. 2c). The SHG signal in pristine 1L-$MoS_2$ shows a sixfold-rotational symmetric pattern with maxima of SHG intensity along its armchair direction and can be well fitted with Eq. (1), implying the $C_3$-rotational symmetry of 1L-$MoS_2$ samples[28]. While in 1L-$MoS_2$/$SiP_2$ heterostructure, the SHG signal presents an additional twofold component imposed to the six symmetric petals (detailed analysis in "Methods"). Such a twofold component does not originate from the $C_3$-symmetric lattice of 1L-$MoS_2$ itself, but results from the reduced symmetry at the 1L-$MoS_2$/$SiP_2$ heterointerface. The SHG intensities can be well fitted with Eq. (2) in the "Methods", similar to those distorted SHG scenarios in uniaxially-strained 1L-$MoS_2$ samples[29,30]. Note that $SiP_2$ itself has no SHG signal due to the existing inversion symmetry in its crystal lattice (Supplementary Fig. 9c), ensuring that the observed SHG signal of 1L-$MoS_2$/$SiP_2$ heterostructure mainly originates from 1L-$MoS_2$ whose band structure is effectively modified by the potential on the heterointerface created by the bottom $C_2$ symmetric $SiP_2$. More importantly, the anisotropic SHG response at 1L-$MoS_2$/$SiP_2$ changes little with increasing $SiP_2$ thickness (Supplementary Fig. 11), further confirming that this is an interfacial phenomenon induced by symmetry breaking at the 1L-$MoS_2$/$SiP_2$ heterointerface between the 1L-$MoS_2$ and the topmost $SiP_2$ layer.

To investigate the effect of symmetry engineering on the optical properties of such an interface, we performed polarization-dependent photoluminescence (PL) measurements in the 1L-$MoS_2$/$SiP_2$ heterostructure at 77 K (Fig. 2d, e and Supplementary Fig. 6). For $SiP_2$, the PL signal with an excitonic emission energy of 2.06 eV at 77 K shows a linear polarization along the $x$ direction of the lattice[26]. In contrast, for 1L-$MoS_2$ without $SiP_2$ stacking layer, the excitonic emission of 1L-$MoS_2$ at 1.91 eV at 77 K remains unchanged with the detection polarization angle and shows no linear polarization. While, for the 1L-$MoS_2$/$SiP_2$ heterostructure, the excitonic state of 1L-$MoS_2$ with an emission energy of 1.91 eV becomes linearly polarized along the $x$ direction of $SiP_2$ (Fig. 2f and Supplementary Fig. 7). The similar results can be observed at 77 K and 300 K in 1L-$WS_2$/$SiP_2$ heterostructure (Supplementary Fig. 8), indicating the anisotropic PL response in TMDC/$SiP_2$ is robust with temperatures. Similar anisotropic SHG responses are also observed in 1L-$WS_2$/$SiP_2$ heterostructure (details in Supplementary Fig. 9), indicating that $SiP_2$ dielectric can effectively engineer the symmetry of its neighboring monolayer TMDC through tunable interlayer interactions. More interestingly, the anisotropic PL and SHG responses strongly depended on the twist angle between $MoS_2$ and $SiP_2$, and the corresponding anisotropy dramatically decreased when the mirror symmetry of moiré superlattice varnishes with changing the twist angle (Supplementary Figs. 12 and 13). This result indicates that the mirror symmetry of $MoS_2$/$SiP_2$ moiré superlattice plays an important role in controlling the magnitude of anisotropic optical responses at the heterointerface. The symmetry breaking induced anisotropic behavior can exist at those interfaces stacked with the $C_3$-symmetric monolayer TMDCs and $C_2$-symmetric dielectrics, enabling a strategy to explore applications such as polarization-sensitive photodetectors[31].

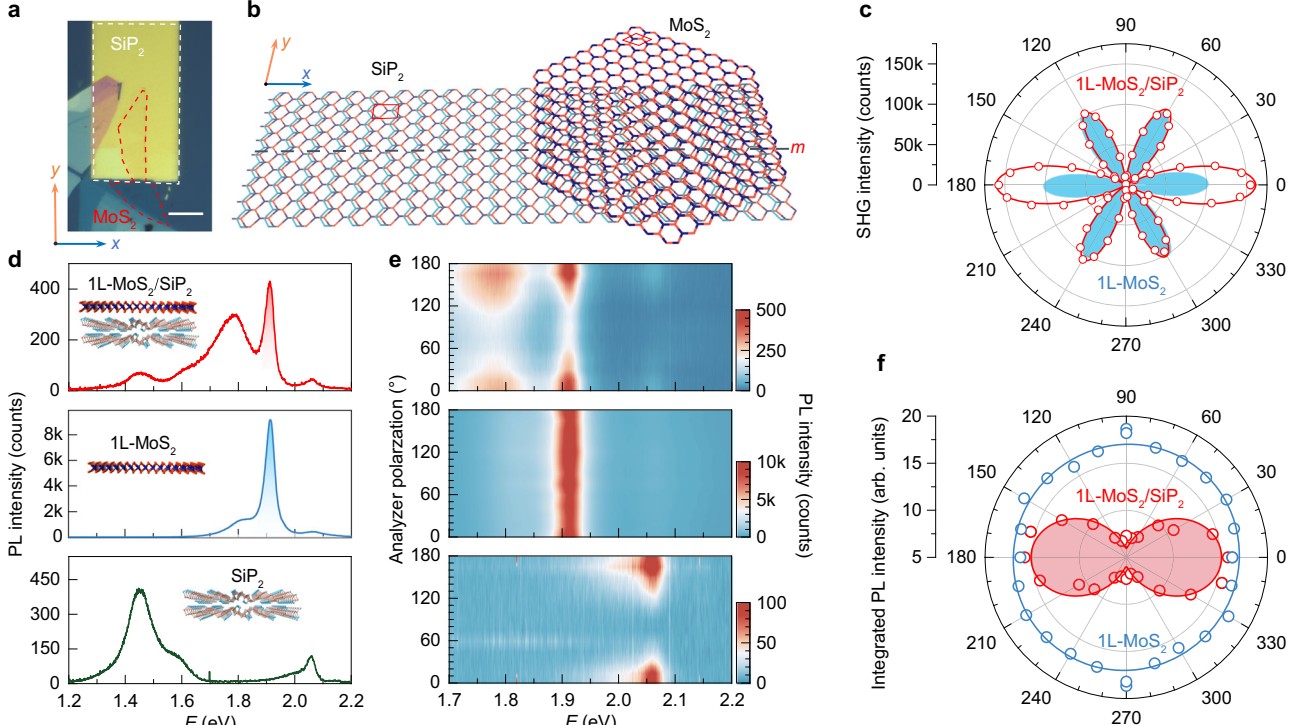

**Fig. 2 | Anisotropic optical response at the TMDC/SiP$_2$ interface. a, b** Optical image (**a**) and schematic illustration of the top view (**b**) of a 1L-MoS$_2$/SiP$_2$ heterostructure, in which the armchair direction of MoS$_2$ and the $x$ direction of SiP$_2$ are parallel. The red and white dashed lines in (**a**) highlight the 1L-MoS$_2$ and SiP$_2$ sample areas. Scale bar is 20 μm. The red rectangle and diamond in (**b**) represent the unit cells of SiP$_2$ and MoS$_2$, respectively. The black dashed line represents the mirror plane $m$. **c** Polar plot of polarization-resolved second harmonic generation (SHG) intensities of bare 1L-MoS$_2$ (blue shadow) and 1L-MoS$_2$/SiP$_2$ (red) under the parallel configuration (the detection polarization is parallel to the excitation polarization). The red solid line represents the fitting with Eq. (2) in the "Methods".

**d** Photoluminescence (PL) spectra of SiP$_2$ (green curve), 1L-MoS$_2$ (blue curve), and 1L-MoS$_2$/SiP$_2$ heterostructure (red curve) at 77 K. Insets are the corresponding schematics of the measured sample geometry. The exciton emission of 1L-MoS$_2$ is highlighted with shadows. **e** Corresponding color plot of the PL intensity in (**d**) as a function of emission photon energy at different detection polarization angles $\theta$. Here $\theta$ denotes the angle between the analyzer polarization direction and the $x$ direction, as defined in (**b**). **f** Polar plots of polarization-resolved PL integrated intensities of 1L-MoS$_2$ (blue curve) and 1L-MoS$_2$/SiP$_2$ (red curve and red shaded area). The circles represent the experimental data and the solid curves represent the sine fitting results.

## Giant anisotropic conductance in SiP$_2$-gated MoS$_2$ transistors

To investigate the interfacial symmetry modulation on the electronic transport properties of MoS$_2$, we measured the conductance $G_x$ ($G_y$) along the $x$ ($y$) directions of SiP$_2$-gated 1L-MoS$_2$ transistors (Fig. 3a, Supplementary Figs. 14 and 15). Figure 3b shows a comparison between $G_x$ and $G_y$ under different $V_{tg-SiP_2}$. One can see that the anisotropy index $G_y/G_x$ can be as high as $10^3$ at the off-state with $V_{tg-SiP_2} < 1$ V (Fig. 3c), implying that the symmetry engineering using SiP$_2$ dielectric can drive the isotropic conductivity of $C_3$-symmetric 1L-MoS$_2$ into highly-anisotropic electronic states. With further increasing $V_{tg-SiP_2}$, the anisotropy index gradually approaches the value of 1, suggesting that 1L-MoS$_2$ recovers back to isotopically conducting states at the on-state. Such continuous modulation of $G_x$, $G_y$, and $G_y/G_x$ index can also be achieved at a wide range of $V_{bg-SiO_2}$ (Fig. 3d–f). The tunable conductance from anisotropic to isotropic characteristics suggests that SiP$_2$ with in-plane anisotropy is anticipated to stimulate device functionality exploration for anisotropic digital inverters[32], anisotropic memorizers[33], or artificial synaptic devices[34].

To confirm that such anisotropic conductance originates from the MoS$_2$/SiP$_2$ heterointerface, we compared the $G_x$ and $G_y$ values of SiP$_2$-gated MoS$_2$ transistors by increasing the thickness of MoS$_2$ from monolayer to 20 nm. As a result, the observed anisotropy index $G_y/G_x$ at the off-state decreases rapidly to ~1 when the thickness of MoS$_2$ is increased to 20 nm (Fig. 3g, Supplementary Figs. 16 and 17), suggesting a nearly isotropic conductance in thicker samples. Such a thickness-dependent behavior is proposed to be attributed to the competition

between the surface and bulk conductance, as qualitatively described in Supplementary Fig. 18. Specifically, the anisotropy index of MoS$_2$ can be written as $\frac{G_y}{G_x} = \frac{G_y^{surface} + G_y^{bulk}}{G_x^{surface} + G_x^{bulk}}$, where bulk conductance (proportional to the sample thickness) is isotropic $G_y^{bulk} \approx G_x^{bulk}$ while surface conductance is anisotropic since only the surface layer of MoS$_2$ with certain thickness (within the Thomas-Fermi screening length) can be tuned for carrier accumulation with SiP$_2$ dielectric based on our numerical Poisson-Schrödinger calculations (Supplementary Figs. 19 and 20). For SiP$_2$-gated 1L-MoS$_2$ case, $G_y^{bulk} = G_x^{bulk} = 0$ and only the MoS$_2$ layer on the interface (namely the whole monolayer) contributes to the conductance, so the anisotropy index can be written as $\frac{G_y}{G_x} = \frac{G_y^{surface}}{G_x^{surface}}$, whose value can be as high as 1000. When increasing the thickness of MoS$_2$, $G_y^{bulk}$ and $G_x^{bulk}$ begin to increase and gradually dominate the total conductance with $G^{surface} \ll G^{bulk}$ at the thick limit. As a result, the anisotropy index is reduced to $\frac{G_y}{G_x} = \frac{G_y^{bulk}}{G_x^{bulk}} = 1$, which is consistent with our experimental observation of less anisotropy in SiP$_2$-gated MoS$_2$ with a thickness of 20 nm. This result indicates that the anisotropic conductance behavior is contributed exactly from the interface of the MoS$_2$/SiP$_2$ heterostructures (details in Supplementary Figs. 18–20). Compared to those vdW materials with in-plane anisotropic crystal lattices and electronic structures[18,35], our SiP$_2$-gated 1L-MoS$_2$ not only has the largest anisotropy index but also hosts the greatest capability to tune such an anisotropy index (Fig. 3h).

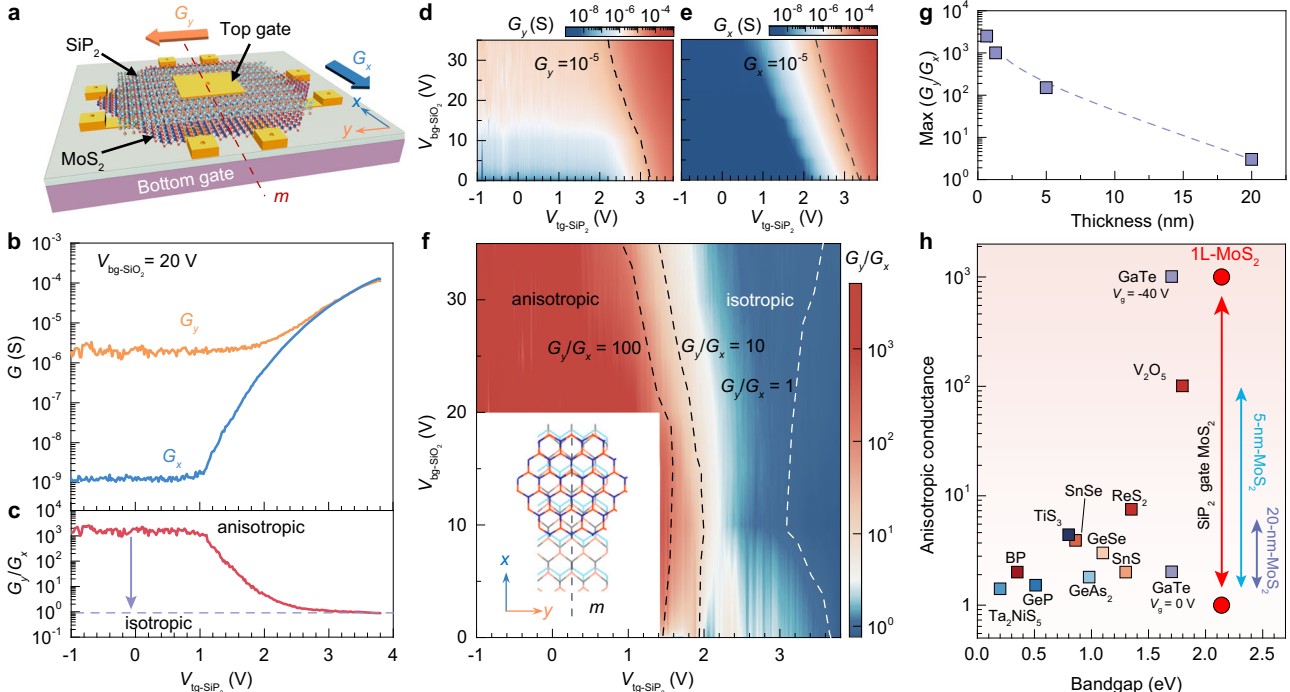

**Fig. 3 | Gate tunable anisotropic transfer characteristics of MoS₂/SiP₂ hetero-interfaces. a** Schematic diagram of a SiP₂-gated MoS₂ transistor. The top gate medium is 20-nm-thick SiP₂, and the bottom gate medium is 300-nm-thick SiO₂. $G_x$ and $G_y$ are the sheet conductance along the $x$ and $y$ directions of the hetero-structure, corresponding to the measurement geometries described in Supplementary Fig. 10. **b** Transfer characteristics of $G_x$ and $G_y$ for 1L-MoS₂ at $V_{bg-SiO_2}$ = 20 V and 2 K. Note that $V_{bg-SiO_2}$ is necessarily applied to reduce contact resistance and activate top-gated channel for achieving measurable four-terminal conductance and a good signal-to-noise ratio. **c** Anisotropy index $G_y/G_x$ as a function of $V_{tg-SiP_2}$ at $V_{bg-SiO_2}$ = 20 V. **d, e** $G_y$ (**d**) and $G_x$ (**e**) as a function of $V_{tg-SiP_2}$ and $V_{bg-SiO_2}$. **f** Color plot of anisotropy index $G_y/G_x$ as a function of $V_{tg-SiP_2}$ and

$V_{bg-SiO_2}$. The dashed lines highlight where $G_y/G_x$ equals 1, 10, and 100. $G_y/G_x$ values for $V_{tg-SiP_2}$ < 1.4 V and $V_{bg-SiO_2}$ < 20 V are not given since $G_x$ is too small to achieve measurable four-terminal conductance. The inset is a schematic illustration of the top view of a 1L-MoS₂/SiP₂ heterostructure. **g** The maximum $G_y/G_x$ value of the MoS₂/SiP₂ device as a function of MoS₂ thickness. **h** The anisotropic conductance in layered materials as a function of their bandgap values. Note that all other aniso-tropic materials host intrinsically-anisotropic conductance in nature while in our case we can drive the intrinsically-isotropic conductance in MoS₂ into the aniso-tropic state. The values of anisotropic conductance and bandgaps for other ani-sotropic materials are generated from previous reports[18,35].

## Anisotropic moiré potential at 1L-MoS₂/SiP₂ interface

To further understand the influence of interface structures on the anisotropic optical and transport behavior, we explore the structural and electronic properties of the 1L-MoS₂/SiP₂ heterostructure by using density functional theory (DFT) calculations. Note that the mirror symmetry of the constructed MoS₂/SiP₂ heterointerface originates from the parallel or antiparallel alignment of the zigzag chain in MoS₂ along the $y$ direction of SiP₂ (see details in Supplementary Fig. 21). Thus, two kinds of moiré patterns can be obtained (labeled as case-I and case-II, details in Section 13 in Supplementary Information). Taking the moiré pattern of case-I as an example (Fig. 4a, b), due to the lattice mismatch between MoS₂ and SiP₂, there are three typical stacking structures, labeled as I-AA, I-AB, and I-BA, as indicated by the colored rectangular areas in Fig. 4a. The details about atomic stacked config-urations and stacked structures with II-AA, II-AB, and II-BA in the moiré pattern of case-II are shown in Supplementary Fig. 21d.

The structural corrugations in 1L-MoS₂/SiP₂ heterostructure with two kinds of moiré patterns are simulated via DFT calculations (Fig. 4c and Supplementary Fig. 27b, e). After being placed on the SiP₂ lattice, the atomic flat structure of 1L-MoS₂ will be deformed due to the interface coupling, resulting in the formation of moiré potential on 1L-MoS₂ that effectively breaks $C_3$-rotational symmetry of the pristine MoS₂. In contrast, the structural corrugations in 1L-SiP₂ are much smaller compared with that in 1L-MoS₂ (Supplementary Fig. 25). Fur-thermore, we plot the distribution of the interlayer distance (marked in Fig. 4b) for the moiré pattern of case-I in Fig. 4c to demonstrate the moiré potential in this heterostructure. In the moiré pattern of case-I, the stacked region of I-BA hosts the smallest interlayer distance

between MoS₂ and SiP₂, indicating the largest structural deformation on 1L-MoS₂ and interlayer coupling. However, in the moiré pattern of case-II such region with the largest moiré potential and lattice defor-mation becomes II-AA (Supplementary Fig. 27e). On the other hand, the mirror symmetry parallel to the armchair direction of 1L-MoS₂ (also the $x$ direction of the heterostructure) is always observed in both moiré patterns of 1L-MoS₂/SiP₂ heterostructures that is determined by the specific stacked regulations in the fabrication of experimental devices. Such characters with reduced symmetry in corrugated 1L-MoS₂ are consistent with the symmetry analysis to stacking structures with moiré patterns. Our simulated structural deformation of corru-gated 1L-MoS₂ with moiré patterns gives a consistent interpretation on the change of the symmetric shape of the experimental SHG spectra from sixfold (1L-MoS₂) to twofold (1L-MoS₂/SiP₂).

The out-of-plane structural corrugations of 1L-MoS₂ with moiré patterns can strongly modulate its electronic structures and thus influence the optical properties. Since the direct simulation of the large-size moiré lattice by using DFT are too expensive to afford, to overcome this issue, we use a strained MoS₂ and SiP₂ to construct a heterostructure guaranteeing the conduction band offset between MoS₂ and SiP₂ same with that in the moiré heterostructure, then simulate the influence of the moiré potential on the electronic states of 1L-MoS₂ (details in Sections 14–17 in Supplementary Information). The exemplified results are presented in the heterostructure model with stacking regions of I-AB and I-BA (named as case-I-ABBA). Figure 4g and h demonstrates the charge density distribution for conduction band edge in case-I-ABBA, and Fig. 4e and f shows the plane-averaged charge density along the $z$ and $y$ directions (Section 17 in

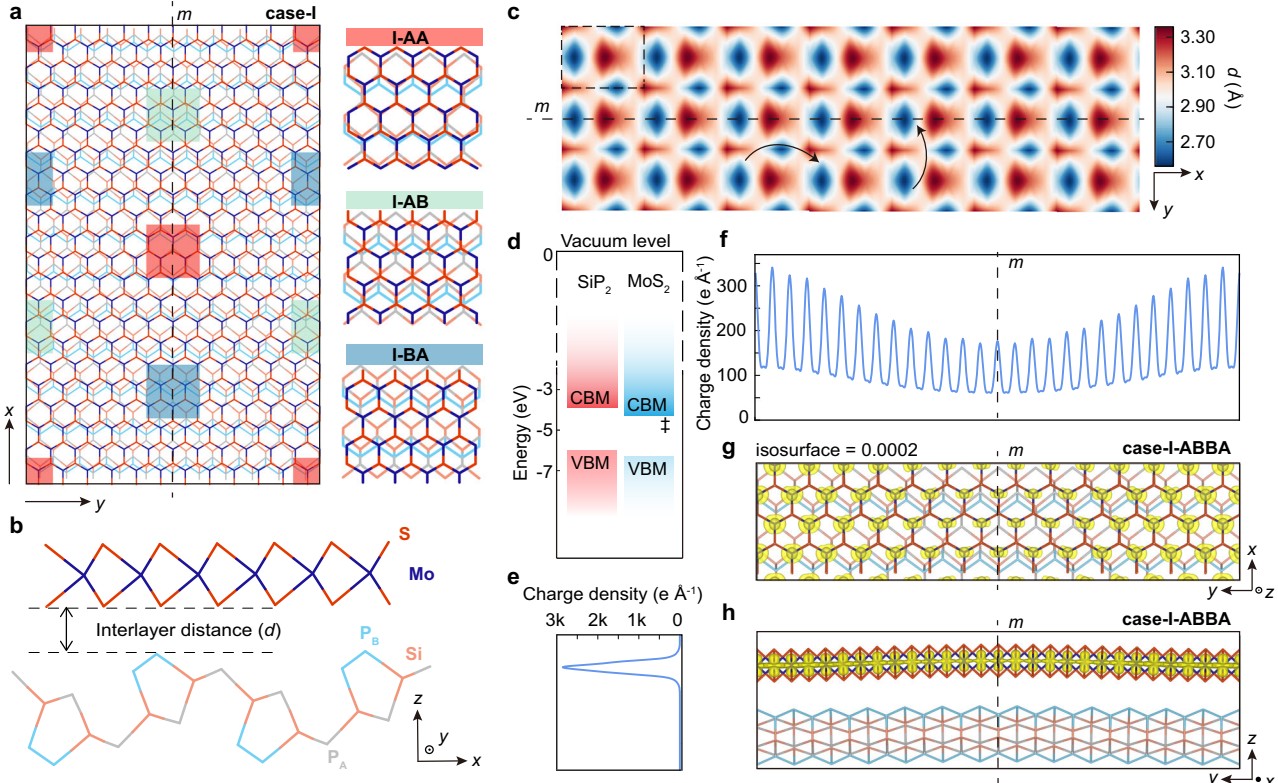

**Fig. 4 | Structural and electronic properties of the MoS₂/SiP₂ heterointerface.**
**a** Moiré pattern of case-I formed by stacking MoS₂ monolayer on SiP₂ monolayer. The colored rectangles mark different stacked configurations: I-AA (red area), I-AB (green area), and I-BA (blue area). The vertical dashed line represents the mirror plane $m$. **b** The side view of the case-I heterostructure model. The interlayer distance is marked by the black lines, which corresponds to the difference between the $z$ coordinates of adjacent S atoms and P atoms in the vdW gap. **c** The real space distribution of the interlayer distance in the moiré pattern of case-I. The dashed rectangular area corresponds to the moiré superlattice shown in (**a**). The effective hopping paths between the trapped states along the $x$ and $y$ directions are shown by the black curve arrows. **d** The band alignment between unstrained monolayer MoS₂ and unstrained monolayer SiP₂. CBM and VBM represent the conduction band minimum and valence band maximum, respectively. The work function and bandgap for each layer are calculated with the HSE-06 functional. The conduction band edge contributed by monolayer MoS₂ is marked by "‡". **e**, **f** The illustrations of the plane-averaged charge density on the lowest conduction band edge contributed by 1L-MoS₂ along the $z$ direction (**e**) and $y$ direction (**f**) of the case-I-ABBA heterostructure. **g**, **h** Top view (**g**) and side view (**h**) for the calculated charge density for the lowest conduction band edge contributed by monolayer MoS₂ in case-I-ABBA heterostructure. The isosurface is set as 0.0002 e Å⁻³.

Supplementary Information). In the heterostructure model with fully relaxation, out-of-plane corrugation can be clearly observed with the retaining of the mirror symmetry in the moiré pattern. The conduction band edge in 1L-MoS₂/SiP₂ heterostructure is dominated by the state from the MoS₂ layer, while its charge density distribution has been strongly modified by the lattice deformation. The calculated results for other heterostructure models containing different stacked regions are shown in Section 17 in Supplementary Information. The conduction band edge in 1L-MoS₂/SiP₂ heterostructure is always strongly modified by the moiré potential. For the case-II moiré pattern, we build similar models and obtain the same conclusions (details in Sections 16 and 17 in Supplementary Information). Compared with pristine 1L-MoS₂ with $C_3$-rotational symmetry, the symmetry engineering on the conduction band edge can be clearly observed in 1L-MoS₂/SiP₂ heterostructure. With breaking $C_3$-rotational symmetry by SiP₂, the conduction band edge on deformed 1L-MoS₂ only keeps the mirror symmetry (Fig. 4f). Therefore, when one electron is excited on conduction band edge and couples with hole states, the optical matrix elements in formed exciton should be strongly modified by the moiré potential with lower symmetry and the lowest bright exciton absorption becomes highly anisotropic, which is consistent with the observation from our PL experiments.

The formation of moiré potential with symmetry engineering can also explain the experimentally observed giant anisotropic conductance in SiP₂-gated MoS₂ transistors. At the off-state with low

carrier density (~5 × 10⁹ cm⁻²) and low temperature (2 K), the charge carriers in 1L-MoS₂/SiP₂ heterostructure are mainly trapped by charged impurities[22,36–38] and the moiré potentials, thus the giant anisotropic conductance of 1L-MoS₂ is mainly contributed by the effective hopping between trapped charge states in the moiré potentials[22,36–38]. Since the charged defects in 1L-MoS₂ are distributed randomly without anisotropy, the anisotropic moiré potential should be a critical factor for the anisotropic conductance in the SiP₂-gated MoS₂ transistor. Similar to previous discussions, we also take the moiré pattern of case-I as an example (the discussion of the moiré pattern of case-II draws the same conclusion), the distribution of interlayer distance between SiP₂ and MoS₂ in real space (Fig. 4c) shows that the smallest interlayer distance is located in the I-BA stacking region, which corresponds to the largest interlayer potential and can effectively trap the charge carriers inside. On the other hand, the anisotropic moiré potential results in highly anisotropic hopping between trapped states. For example, the effective hopping along the parallel direction (∥) to the mirror plane is naturally smaller than that along the perpendicular direction (⊥), indicating that, at the off state, the effective mass of these trapped states is highly anisotropic ($m_∥ \gg m_⊥$). The large ratio of effective mass ($m_∥/m_⊥$) leads to the highly-anisotropic conductance in 1L-MoS₂/SiP₂ heterostructure. With increasing the electron density to reach the on-state, the moiré potential cannot fully trap these charge carriers and its influence on transport becomes unimportant, delocalizing the 2D electron gas

formed on the 1L-MoS$_2$ layer. Thus, the conductance turns out to be isotropic at the on-state.

## Discussion

In conclusion, we demonstrate an anisotropic van der Waals dielectric SiP$_2$ that can simultaneously tune the electronic states of channel semiconductors and induce symmetry engineering at TMDC/SiP$_2$ interfaces. Our first-principles calculations reveal that these anisotropic characteristics originate from the formation of the anisotropic-symmetric moiré potential in the MoS$_2$/SiP$_2$ heterostructure, which strongly modulates structural and electronic properties of 1L-MoS$_2$ with tunable anisotropic symmetry. The tunable interfacial symmetry in the TMDC/SiP$_2$ heterostructure can provide a unique platform for investigating symmetry-related interfacial physics and corresponding phenomena, including the generation of the in-plane polarization[5] (bulk photovoltaic effect and quantum shift current) and the Berry curvature dipole[39–41] (circular photo-galvanic effect and nonlinear Hall effect). The giant anisotropy generated in the TMDC/SiP$_2$ heterostructure, which is absent in the pristine TMDC material, sheds light on the moiré physics of the engineered interface with reduced symmetry, and provides an effective way to control the degree of freedom of electrons in condensed matter systems.

## Methods

### Crystal symmetry analyses of TMDC/SiP$_2$ heterostructures

The orthorhombic SiP$_2$ crystal exhibits an anisotropic layered structure (space group *Pnma*) with an embedded quasi-one-dimensional P$_B$–P$_B$ chains along the $y$ direction of the crystal lattice. Specifically, three important spatial symmetry operations should be addressed in this atomic structure of the SiP$_2$ crystal when stacking SiP$_2$ with monolayer TMDCs. First, there is a non-symmorphic $C_2$ symmetry about the $z$ direction (the screw symmetry $S_2$ that combines a twofold rotational symmetry with a translation along the $z$ direction in the half-unit-cell, $S_2 = C_2 + z/2$) in the SiP$_2$ crystal. This non-symmorphic $C_2$ symmetry of bulk SiP$_2$ is incompatible with $C_3$ symmetry of TMDCs and will result in highly anisotropic nature ($C_1$ symmetry) of the TMDC/SiP$_2$ heterostructures. Second, the atomic structure of SiP$_2$ is inversion symmetric, which means there is no SHG signal of SiP$_2$ flakes, ensuring that the distorted SHG signals at the TMDC/SiP$_2$ heterostructure mainly come from the symmetry breaking at the heterointerface. Third, there is a mirror symmetry perpendicular to the P$_B$–P$_B$ chains ($y$ direction) of the SiP$_2$ crystal. This vertical mirror can remain in the TMDC/SiP$_2$ heterostructures when stacking the TMDCs and SiP$_2$ by aligning their mirror planes (the zigzag direction of TMDCs is parallel to the P$_B$–P$_B$ chains of SiP$_2$), generating a mirror symmetric anisotropic moiré potential at the heterointerface.

### Optical measurements of TMDC/SiP$_2$ heterointerfaces

SiP$_2$ flakes and TMDC flakes were prepared by mechanical exfoliation onto polydimethylsiloxane (PDMS) stamps and SiO$_2$/Si wafers (300-nm-thick SiO$_2$ layer). TMDC/SiP$_2$ heterointerfaces were fabricated using a dry-transfer method and stacked by parallelly aligning the zigzag direction of MoS$_2$ and $y$ direction of SiP$_2$. The crystal axes of the 1L-MoS$_2$ samples are confirmed by their polarized-SHG results. And the crystal axes of SiP$_2$ are first identified by their optical image and then confirmed by their polarized PL results. The whole sample fabrication was processed in a glove box to avoid any degradation. Room temperature PL measurements were performed using a confocal Raman system (WITec Alpha 300) using a 50× objective lens with an incident laser (laser power of 1 mW) focused to a 1 μm spot size. Nitrogen-filled environments were established by protecting samples with continuous nitrogen gas flow. Low-temperature PL measurements were performed under vacuum conditions in cryostats (Cryo Instrument of America RC102–CFM Microscopy Cryostat). For polarized PL measurements, the excitation polarization is fixed along the $x$ direction, and the

detection polarization is changed from $\theta = 0°$ to $180°$ ($\theta$ is the angle between the detection polarization and the $x$ direction).

The SHG measurements were performed using a Ti:sapphire oscillator with an excitation wavelength of 810 nm, pulse width of 70 fs, and repetition rate of 80 MHz. The laser pulse was focused to an ~1 μm spot size by a 40× objective lens. The SHG signals are obtained under a configuration with the detection polarization parallel to the excitation polarization. For the SHG signal of 1L-TMDCs on the SiO$_2$ substrate, the sixfold symmetric SHG intensities are fitted by Eq. (1):

$$I_{\text{SHG}}^{\parallel} \propto \cos^2 3\theta \tag{1}$$

Normally, for those uniaxially-strained TMDCs, the SHG intensities $I_{\text{SHG}}^{\parallel}$ parallel to the incident laser polarization can be written as:

$$I_{\text{SHG}}^{\parallel} \propto \left[ \cos 3\theta + \varepsilon_y \left( k_1 \cos^3 \theta - k_2 \sin^2 \theta \cos \theta \right) \right]^2 \tag{2}$$

where $\varepsilon_y$ is the strain along the $y$ direction, $k_1$ and $k_2$ are parameters related to TMDCs. In our case of the 1L-MoS$_2$/SiP$_2$, the $C_3$ symmetry of 1L-MoS$_2$ is also reduced to low symmetry such as $C_1$. Therefore, we fit our data by Eq. (2) to describe the anisotropic SHG response in our 1L-MoS$_2$/SiP$_2$.

### Electrical transport measurements

The 1L-MoS$_2$ (or few-layer MoS$_2$) and SiP$_2$ flakes for electronic transport measurements were exfoliated onto a PDMS stamp and transferred onto a silicon substrate with prepatterned electrodes (Ti/Au with a thickness of 3/9 nm) in sequence. A top gate on the SiP$_2$ flake (Ti/Au with a thickness of 5/45 nm) was then made using electron-beam lithography and electron-beam evaporation. Electrical transport measurements were performed in a cryo-free superconducting magnet system (Oxford Instruments Teslatron$^{\text{PT}}$). Four-terminal resistance $R_{xx}$ was acquired using a Keithley 2182 voltmeter with a DC current supplied by a Keithley 2400 sourcemeter. The gate voltage is supplied by a Keithley 2400 sourcemeter. The sheet carrier density ($n_{2D}$) is obtained based on Hall effect measurements on Au/SiP$_2$/MoS$_2$ sandwiched devices[22]. The Au/SiP$_2$/MoS$_2$ device can be considered as a parallel plate capacitor, and the amount of charge per unit area can be written as:

$$e n_{2D} = \frac{\varepsilon_0 \varepsilon_{r-SiP_2}}{t_{SiP_2}} V_{tg-SiP_2} \tag{3}$$

where $e$ is the electron charge, $\varepsilon_0$ is the vacuum permittivity, $t_{SiP_2} = 20$ nm is the thickness of SiP$_2$, and $\varepsilon_{r-SiP_2}$ is the relative dielectric constant of SiP$_2$ within the Au/SiP$_2$/MoS$_2$ sandwiched structure. The $\varepsilon_{r-SiP_2}$ is obtained by linear fitting of Eq. (3).

### DFT calculations

The Vienna Ab initio Simulation package (VASP)[42] was used for the first-principles calculations. The generalized-gradient approximation (GGA) of the Perdew-Burke-Ernzerhof-type functional[43] was used with the projected-augmented-wave method[44,45] and an energy cutoff of 500 eV. For the electronic self-consistent calculations, the convergence criterion was set as $10^{-6}$ eV. Considering the van der Waals interaction, the DFT-D3 method[46] was applied as the correction. The force criterion was chosen to be 0.01 eV Å$^{-1}$. After fully relaxing the 2H-MoS$_2$ bulk structure, we obtained the lattice constants of 3.16 Å for $a$ (and $b$) and 12.39 Å for $c$, which were consistent with previous studies[47,48]. The $k$-point mesh of $15 \times 15 \times 3$ was used to sample the Brillouin zone (BZ). After the full relaxation of the SiP$_2$ bulk structure, the lattice constants were 10.11 Å for $a$, 3.44 Å for $b$, and 14.18 Å for $c$, which were consistent with the experimental results[49]. The $k$-point mesh of $5 \times 15 \times 4$ was used to sample the BZ of bulk SiP$_2$. Moreover,

except for the structural relaxation, the spin-orbit coupling was considered in all DFT calculations. For all slab models used in this work, a vacuum layer with 20 Å was added along the $z$ direction. Moreover, the calculations with the HSE06 functional[50,51] were performed to obtain the accurate values of the work function and bandgap of the unstrained and strained $MoS_2$ and $SiP_2$ monolayers.

For these heterostructure models with strained $MoS_2$ and $SiP_2$ which were used to simulate the electronic properties of 1L-$MoS_2$/$SiP_2$ heterostructure directly, the $2\sqrt{3} \times 1$ rectangle supercell for monolayer $MoS_2$ was used and its lattice was tensed to 3.25 Å for $a$, and was compressed to 10.69 Å for $b$. Correspondingly, the lattice of monolayer $SiP_2$ was enlarged to 3.50 Å for $a$ and 10.69 Å for $b$. In each strained heterostructure model, it contains $13 \times 1$ monolayer $SiP_2$ unitcells and $14 \times 1$ rectangle supercells for monolayer $MoS_2$. The $k$-point mesh of $5 \times 1 \times 1$ was used to sample the BZ of heterostructure model. For strained case-I-ABBA and case-II-AABA lattices, we fully relaxed the whole structures. While, for strained case-I-AA and case-II-AB lattices, we fixed the $x$ and $y$ coordinates of the two central S atoms in $MoS_2$ layer and one central P atom in $SiP_2$ layer in the AA/AB stacking region. Then we fully relaxed other atoms in the strained heterostructure model.

## Data availability
The Source Data underlying the figures of this study are available at https://doi.org/10.6084/m9.figshare.23623722. All raw data generated during the current study are available from the corresponding authors upon request.

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

## Acknowledgements

This work was supported by the A3 Foresight Program—Emerging Materials Innovation. The authors would like to acknowledge the support by the National Natural Science Foundation of China (grant nos. 51861145201 (H.T.Y.), 52072168 (H.T.Y.), 21733001 (H.T.Y.), 12204232 (F.Q.), 12234011 (P.T.)), the National Key Research and Development Program of China (grant nos. 2018YFA0306200 (H.T.Y.), 2021YFA1202901 (J.H.)), the Natural Science Foundation of Jiangsu Province (grant no. BK20220758 (F.Q.)), KAKENHI grant JP19H05602 (Y.I.) and JP23H00088 (T.I.) from Japan Society for the Promotion of Science (JSPS) and JST FOREST (grant no. JPMJFR213A (T.I.)).

## Author contributions

Z.L., J.H., L.Z., and Z.X. equally contributed to this work. H.T.Y., P.T., and Y.I. conceived the project and designed the experiments. Z.L., X.S., and C.Q. fabricated samples for optical measurements and carried out PL measurements. L.Z., Z.L., P.C., and C.S. fabricated transport devices. J.H. and L.Z. carried out electrical resistance measurements. G.L. and X.X. carried out SHG measurements. Y.L. and H.G. synthesized the $SiP_2$ crystals. L.Z. and Z.L. performed Poisson-Schrödinger equation calculations. Z.X. and P.T. performed ab initio calculations and model simulations. Z.L., F.Q., T.I., and Y.I. analyzed the data. Z.L. and H.T.Y. wrote the manuscript with all the authors' input.

## Competing interests

The authors declare no competing interests.
