## [Peer review file · Nature Communications]

REVIEWER COMMENTS

Reviewer #1 (Remarks to the Author):

The authors reported some interesting results on using the different symmetry in SiP2 dielectric and MoS2 semiconductor channels to induce strongly anisotropic electric/optical effects in otherwise isotropic MoS2 2D semiconductor devices. This is a novel idea and the tunability of anisotropy by gate will attract interest from the community. The experimental work is also coupled with DFT calculations of the bandstructure change to explain the data.

Before recommending the paper for publication, the authors should discuss/clarify the following points. (1) In addition to the anisotropic 2D materials mentioned, there are also 2D oxides with strongly anisotropic inplane structure, yielding > 100-time anisotropic conductance inplane, see, Sukrit Sucharitakul et al, "V2O5: a 2D van der Waals Oxide with Strong In-plane Electrical and Optical Anisotropy", ACS Appl. Mater. Interfaces, 9 (28), 23949–23956 (2017). This category of 2D materials shouldn't be left out and actually falls in the open area between this work and other anisotropic 2D materials which normally have anisotropic factor $\sim 1-10$ (shown in fig.3h of the paper). (2) Anisotropic conductance and anisotropic conductivity are two different things. Depending on the measurement geometry, the anisotropic conductance measured can be greatly enhanced compared to the conductivity. See ACS Appl. Mater. Interfaces, 9 (28), 23949–23956 (2017) which has a detailed discussion/analysis. (e.g. if the current is injected into the sample by point-like contacts, then 1000 times anisotropy in conductance actually corresponds to $\sim 20-30$ times anisotropy in conductivity because of the non-uniform current flow effect.) I don't think this will reduce the significance/impact of this work but should be discussed and clarified to improve the rigorosity of the paper.

Reviewer #2 (Remarks to the Author):

In this manuscript, Li et al. reported a new non-symmetric dielectric material, SiP2, for inducing anisotropic electrical conductivity, linearly-polarized PL and anisotropic SHG in 2D MoS2 that has a symmetric C3 structure. They attributed these phenomena to the anisotropic Moiré potential at the SiP2/TMDC interfaces. Such an idea of utilizing interface symmetry engineering on 2D heterostructures and related field-effect transistor (FET) devices for emergent phenomena, which is well-demonstrated in this work, is novel and inspiring for the study of condensed matter physics and advanced electronic applications.

In general, this paper is well organized and well written. To the extent that I can appreciate, it is a novel strategy to adopt such a non-symmetric gate dielectric material for inducing quite some interesting emergent interfacial phenomena. For example, since the TMDC is a well-known material with giant Berry curvature at K valleys, a modulation of the band structure of TMDC from the crystal field of non-symmetric SiP2 dielectric can cause the Berry curvature dipole and other effects governed by the tunable interfacial symmetry. Thus, interesting interfacial physics and corresponding phenomena can be expected by controlling and breaking the symmetry of the interface, including the generation of the in-plane polarization (photovoltaic effect and quantum shift current), the Berry curvature dipole (circular photo-galvanic effect and nonlinear anomalous Hall effect) and the gauge field (Landau level quantization). Since those nonlinear properties usually only exist in those low-symmetric interfaces, the report of such an anisotropic dielectric can provide a new strategy for generating exotic functionalities at dielectric/semiconductor interfaces and related FET devices via interfacial symmetry engineering.

In short, this is a fairly-good manuscript with high quality and broad interest, which would give a substantial contribution to interface physics and twistrionics as well as advanced electronics. I would like to recommend its publication in Nature Communications after clarifying several concerns below.

1. The authors show the generation of giant anisotropy in SiP2/TMDC interfaces by breaking the C3 rotation symmetry of 1L-MoS2 down to C1 using a SiP2 anisotropic dielectric with C2 rotation symmetry. This is very interesting. In principle, there can be in-plane polarization in such a C1 interface (Science 372, 68-72 (2021)). The bulk photovoltaic effect is expected to occur at this SiP2/TMDC dielectric/semiconductor interface. This possibly arises from the shift current mechanism due to the Berry connection at the interface. I suggest the authors discuss such possible exciting phenomena in the prospect part of the manuscript.

2. Similarly, there is also a possibility of generating the Berry curvature dipole. Since the TMDC is a well-known material with giant Berry curvature at K valleys, a slight modulation to the band structure can cause the Berry dipole (Nature Physics 14, 900-906 (2018)). In addition to the anisotropic conductance, one can also expect the nonlinear anomalous Hall effect at this SiP2/TMDC interface due to the symmetry breaking by the SiP2 anisotropic dielectric. The circular photo-galvanic effect is also expected, possibly emerging from the Berry curvature dipole at this SiP2/TMDC interface. Is there any possible experimental sign to confirm that?

3. The band alignment is crucial for the heterostructure, which determines many physical processes, such as charge transfer, band bending effect, and so on. In Fig. 4d, the SiP2/TMDC dielectric/semiconductor interface exhibits type-I band alignment. While in the supplementary, the author shows the type-II band alignment. What is the actual band alignment? I hope the author can experimentally identify the band alignment through Kelvin-probe force microscopy (KPFM) or Ultraviolet photoemission spectroscopy (UPS) measurements.

4. The authors demonstrate a giant anisotropic conductance in the SiP2-gated MoS2 FETs due to the formation of anisotropic moiré potential at SiP2/MoS2 interface via symmetry engineering, which is very interesting. I wonder what is the difference between the results of temperature-dependent conductance/resistance along the two different directions of the SiP2-gated MoS2 FETs?

5. The trion states in 1L-TMDC are widely-observed due to their large binding energies and always provide an ideal platform to study excitonic many-body physics. The authors show that the exciton in 1L-TMDC becomes linearly-polarized in the SiP2/TMDC dielectric/semiconductor interface via symmetry engineering. How about trion states in this system? Does the trion state also show such linear polarization due to symmetry breaking?

Also, I suggest the authors consider addressing the following technical issues:

1. In Fig. 1c and d, the authors show the transfer curves and leakage current of the 5-nm-thick MoS2 FETs. How about the results of the 1L-thick device? And Fig. 1f shows the R_{xx} -T curves tuned by adjusting the SiP2 gate for the 1L-MoS2 FETs. How about the results for thicker samples?

2. In Fig. 1d, the authors label the right axis as "current" for the leakage current of the SiP2-gated MoS2 FETs. However, the unit of right axis is "A cm⁻²". Do the authors want to show the "current density" here? The authors should redefine the label of the right axis precisely.

3. In Fig. 1f, the authors show the insulator-metal transition in SiP2-gated MoS2 FETs. One missing piece of information is the sheet resistivity, R_c (the authors only show four-terminal resistance R_{xx} , which is not universal, while R_c should be) that can be measured accurately — all we need are the sample dimensions — and such a value is much more informative than the resistance R_{xx} .

4. In Fig. 2c, the authors show that the isotropic SHG signal in 1L-MoS2 becomes anisotropic in SiP2/1L-MoS2 heterointerface. Does the thickness of SiP2 influence this phenomenon, such as the anisotropic ratio of the SHG signal? Can the authors discuss the thickness effect of SiP2 on such anisotropic SHG signals?

5. In Fig. 2d, the authors show that the exciton in 1L-MoS₂ becomes linearly-polarized in SiP₂/MoS₂. However, one can see a lower energy peak on the side of the exciton peak in the PL spectra of the SiP₂/1L-MoS₂ heterostructure, which also shows linear polarization. What is this lower energy peak in SiP₂/MoS₂? Is it the interlayer exciton peak?

6. Fig. 2e is the corresponding color plot of the PL intensity in Fig. 2d. In Fig. 2d, the PL intensity of SiP₂ (black curve) and 1L-MoS₂/SiP₂ heterostructure (red curve) appear to be comparable, but in Fig. 2e, why is there a five-fold difference in the intensity color scale between the two?

7. In Fig. 4a, the Moiré period contains 12 hexagons in the y direction, while in Fig. 4g, it contains 14 hexagons. What is the Moiré period (and the size of Moiré Brillouin zone) of this MoS₂/SiP₂ interface?

Reviewer #3 (Remarks to the Author):

Title: A Giant Anisotropic van der Waals Dielectric for Symmetry Engineering in Functionalized Heterointerfaces

Manuscript#: NCOMMS-23-12640

In this manuscript by Z. Li et al., the authors report the realization of anisotropic optical and transport properties in 2D materials by engineering the C₃ rotational symmetry breaking at heterointerfaces. The authors fabricated SiP₂/MoS₂ heterostructure devices, where the SiP₂ layer serves not only as a dielectric gate medium, but also as a C₃ symmetry-breaking factor. Due to the proximal SiP₂ characterized by non-symmorphic C₂ symmetry, the intrinsically isotropic MoS₂ layer evinced linearly-polarized photoluminescence, anisotropic second-harmonic generation, and anisotropic conductance. With additional DFT calculations that suggest the MoS₂ lattice deformation and charge density distribution having mirror symmetry, the authors concluded that the formation of moiré potentials at the SiP₂/MoS₂ interface results in the observed anisotropies.

Engineering rotational symmetry breaking in 2D materials and their heterostructures define a versatile strategy to create emerging optical, electrical, magnetic, and topological properties. Recent examples include engineering uniaxial strain to control the optical responses and magnetism of 2D materials, and forming moiré wires or rectangles to achieve highly-anisotropic transport and optical responses in twisted bilayer systems; this current manuscript by Z. Li et al. made an exciting and unique addition to the progress on engineering rotational symmetry breaking of 2D materials. The C₃ rotational symmetry of 2D materials was broken by a dielectric medium with C₂ symmetry.

While the authors have demonstrated the resultant, highly-anisotropic optical and transport properties in SiP₂/MoS₂, the discussion regarding the fundamental origin of such observations requires further experimental evidence. The authors attributed the origin of the anisotropic conductance index and linearly-polarized optical responses to the formation of moiré superlattices. This argument is inferred from DFT calculations, which suggest the variation of interlayer distances and the charge density distributions having a mirror symmetry due to the modulation by moiré superlattices. The authors argued that the conduction-band-edge charge density distributions imprinted by the moiré result in anisotropic charge (e.g., electrons doped at the band edge) hopping, and the exciton PL polarization. However, no direct experimental evidence supports this argument, e.g., the morphology of the moiré lattice and how the anisotropic properties vary as a function of twist angles.

1. The lack of surface morphology characterizations throughout the current manuscript is a concern. The authors should provide morphological characterizations (e.g., via KPFM, PFM, or STM), demonstrating the formation of anisotropic moiré superlattices at the interface of MoS₂ and SiP₂ with identified twist angles.

2. The symmetry of MoS₂/SiP₂ moiré superlattice, thus, the electronic band structure, optical responses, and electrical properties of the heterostructure depend on the twist angle between MoS₂ and SiP₂. In this regard, the authors should prepare a few MoS₂/SiP₂ devices with varying twist angles, and interrogate the outcome, e.g., the polarization-resolved PL and SHG, and the conductance index.

3. With the particular crystal alignment between MoS₂ and SiP₂, the simulated results, e.g., in Supplementary Figure 20, show visually isotropic moiré-modulated structural corrugations and charge-density distributions, likely having a C₃ rotational symmetry. It is unclear how these visually-isotropic simulated results corroborate the observed anisotropic conductance.

4. To identify the hopping path, the moiré potential should be mapped.

Other comments to clarify:

1. On page 4-5, the authors contrasted the gate voltages for SiO₂ (80V) with that for SiP₂ (5V) when achieving a similar on/off current ratio.

The current should be related to the doping density, and the doping density $n \cdot e$ is generally given by $n \cdot e \approx C_{\text{top}} \cdot V_{\text{tg}} + C_{\text{bot}} \cdot V_{\text{bg}} \approx (\epsilon_{\text{top}} \cdot A / d_{\text{top}}) V_{\text{tg}} + (\epsilon_{\text{bot}} \cdot A / d_{\text{bot}}) V_{\text{bg}}$, where e is the electron charge, C_{top} (C_{bot}) is the capacitance associated with the top (bottom) gate, the top (bottom) dielectric medium, and MoS₂; ϵ_{top} (ϵ_{bot}) is the dielectric constant of the top (bottom) gating dielectric medium; d_{top} (d_{bot}) is the distance between the top (bottom) gate and MoS₂; and V_{tg} (V_{bg}) is top (bottom) gate voltage.

In this regard, the contrasting gate voltages required to achieve the same on/off are naturally determined by the different thicknesses and dielectric constants of the gating medium. It is unclear what the authors were trying to emphasize on page 4-5 when describing the contrasting gate voltages.

2. On page 5, the authors mentioned the mobility of MoS₂ in a SiP₂-gated device vs. that of MoS₂ in an HfO₂-gated device. Since mobility μ is derived from $\mu = e \cdot \tau / (m)$, where e is an electron charge, τ is the carrier scattering time, and m is the carrier's effective mass, why the mobility of MoS₂ depends on the gating medium? Alternatively, does this difference come from device variations?

3. What are the x and y directions in Figure 2a; how were the crystal axes of MoSe₂ and SiP₂ experimentally confirmed in the optical image?

4. How do the SHG anisotropy in Figure 2c and the PL polarization in Figure 2f orient on top of the optical image in Figure 2a? Were the SHG and PL taken from one spot, or multiple spots in the heterostructures? How about the consistency?

5. The microscopic origin of the polarized PL is unclear. The nature of the exciton should be related to the Wannier states confined by the moiré potential (likely at the AA or BA site based on Supplementary Figure 20c and f); how is the optical selection rule derived from this, and how this is related to the PL polarization?

6. If moiré potentials were formed at the interface of MoS₂/SiP₂, the doped charges were likely localized/trapped near the interface of MoS₂/SiP₂ in bulk MoS₂ due the moiré traps; how is the charge wavefunction spatial distribution in the bulk-MoS₂/SiP₂ different from that of the Monolayer-MoS₂/SiP₂, and how does this difference mechanistically give rise to different transport properties?

**Report of Reviewer #1 (Remarks to the Author)**

*The authors reported some interesting results on using the different symmetry in SiP₂ dielectric and*
*MoS₂ semiconductor channels to induce strongly anisotropic electric/optical effects in otherwise*
*isotropic MoS₂ 2D semiconductor devices. This is a novel idea and the tunability of anisotropy by*
*gate will attract interest from the community. The experimental work is also coupled with DFT*
*calculations of the band structure change to explain the data.*

**Authors' response:**

We appreciate the reviewer's positive evaluation and constructive comments on our manuscript,
which greatly helped us improve the quality of the manuscript. We have addressed all the comments
point by point in the revised manuscript, and thus we hope that the revised manuscript meets the
publication criteria of *Nature Communications*.

*Before recommending the paper for publication, the authors should discuss/clarify the following*
*points. (1) In addition to the anisotropic 2D materials mentioned, there are also 2D oxides with*
*strongly anisotropic in-plane structure, yielding > 100-time anisotropic conductance in-plane, see,*
*Sukrit Sucharitakul et al, "V₂O₅: a 2D van der Waals Oxide with Strong In-plane Electrical and*
*Optical Anisotropy", ACS Appl. Mater. Interfaces, 9 (28), 23949–23956 (2017). This category of 2D*
*materials shouldn't be left out and actually falls in the open area between this work and other*
*anisotropic 2D materials which normally have anisotropic factor ~1-10 (shown in fig.3h of the*
*paper).*

**Authors' response:**

We appreciate the reviewer's professional suggestions. We totally agree with the reviewer that those
anisotropic two-dimensional (2D) oxides can host strong in-plane conductance anisotropy¹ (such as
V₂O₅, anisotropic index over 100) which fall in the open area between our work and other anisotropic
2D materials (Fig. 3h in the manuscript). This is indeed a significant addition to the categories of
anisotropic 2D materials for our comparison and should be mentioned in the manuscript.

By following the reviewer's suggestion, we added the data point of 2D oxides (V₂O₅) and replotted
Fig. 3h in the revised manuscript (Fig. R1 here) for a more comprehensive comparison of anisotropic
2D materials. We also cited the reference mentioned by the reviewer accordingly.

**Figure R1 | The anisotropic conductance in layered materials as a function of their bandgap**
**values.** Note that all other anisotropic materials host intrinsically anisotropic conductance in nature,
while in our case, we can drive the intrinsically-isotropic conductance in MoS₂ into the anisotropic
state.

(2) Anisotropic conductance and anisotropic conductivity are two different things. Depending on the
measurement geometry, the anisotropic conductance measured can be greatly enhanced compared
to the conductivity. See ACS Appl. Mater. Interfaces, 9 (28), 23949–23956 (2017) which has a
detailed discussion/analysis. (e.g. if the current is injected into the sample by point-like contacts,
then 1000 times anisotropy in conductance actually corresponds to ~20-30 times anisotropy in
conductivity because of the non-uniform current flow effect.) I don't think this will reduce the
significance/impact of this work but should be discussed and clarified to improve the rigorousness
of the paper.

**Authors' response:**

We appreciate the reviewer's concern on the difference in anisotropic conductance and anisotropic
conductivity. Generally, the real conductance in a material can be significantly influenced by the
measurement geometry and non-uniform current flow effect. However, in order to remove such a
non-uniform current flow effect in our measurements, we use an etched square-shape channel to
strictly ensure that the measurement geometry is equivalent between x and y directions. Hence, the
applied current is uniformly flowing through the channel material and the influence of the
measurement geometry can be neglected in our case.

Please allow us to explain it in detail.

In those V_2O_5 cases using van der Pauw method (Fig. R2a,b), the current injected by point-like
electrodes between E1 and E2 (or E1 and E3) is nonuniform, and the conductance anisotropy is
greatly overestimated compared to its counterpart in conductivity. In contrast, our SiP_2/MoS_2 channel
used in such measurements is intentionally etched into a square-shape to make the current flow
symmetric between x and y directions. We also designed two electrode pads side by side as the source
(or drain) electrode, assuming that these two big electrode pads can serve as one large electrode for
injecting uniform current (Fig. R2c,d). Note that such two special designs can make the current
uniformly flow through the sample channel and ensure that the measurement geometry between x
and y directions is equivalent. Specifically, based on the measurement geometry with current flowing
along x direction of a typical MoS_2/SiP_2 FET (Fig. R2c), we shorted E1 and E2 as the source electrode
(E3 and E4 as the drain electrode) to inject a uniform current across the sample, and we measured
the conductance between the local electrodes E5 and E6 (or between E7 and E8). Figure R2e shows
the conductance between E5 and E6 (or between E7 and E8) as a function of top-gate voltage via
SiP_2 dielectric. One can see that the two different channels exhibit almost the same conductance,
indicating that the current flow through the sample is relatively uniform. We thus believe such a
symmetric measurement geometry can minimize the difference between the measured anisotropic
conductance and the anisotropic conductivity.

To address the reviewer's concern on the difference between anisotropic conductance and anisotropic
conductivity, we added several sentences on page 25 in the revised Supplementary Information:

“Note that the SiP_2/MoS_2 channel used in such measurements is intentionally designed as the square-
shape to make the current flow symmetric between x and y directions. We also designed two electrode
pads side by side as the source (or drain) electrode, assuming these two big electrode pads can serve
as one large electrode for injecting uniform current (Supplementary Fig. 14a,c). Note that such two
special designs can make the current uniformly flow through the sample channel and ensure that
measurement geometry between x and y directions is equivalent. Specifically, based on the
measurement geometry with current flowing along x direction of a typical MoS_2/SiP_2 FET
(Supplementary Fig. 14c), we shorted the E1 and E2 as the drain electrode (E3 and E4 as source
electrode) to inject a uniform current across the sample, and we measured the conductance between
the local electrodes E5 and E6 (or between E7 and E8). Supplementary Fig. 14d shows the
conductance between E5 and E6 (or between E7 and E8) as a function of top-gate voltage via SiP_2
dielectric. One can see that the two different channels exhibit almost the same conductance,
indicating that the current flow through the sample is relatively uniform. We thus believe such a
symmetric measurement geometry can minimize the difference between the measured anisotropic
conductance and the anisotropic conductivity.”

In addition, we have noticed that in our previous manuscript, the conductance is represented by the
 letter “ σ ” (which is usually used for representing the conductivity in other reports^{1,2}). To avoid any
 misleading, we also replaced the letter “ σ ” with the widely-used letter “ G ” to represent the
 conductance throughout the whole revised manuscript.

 **Figure R2 | Evidence of uniform current through the MoS₂/SiP₂ sample. a**, Schematic illustration of the van der Pauw measurement geometry with current flow along y direction. E1 and E2 are used
 as the source-drain electrode, while E3 and E4 are used for measuring the conductance. **b**, Schematic
 illustration of the van der Pauw measurement geometry with current flow along x direction. **c**,
 Schematic illustration of our measurement geometry with current flow along y direction. We short
 the E1 and E2 as the drain electrode while E3 and E4 as the source electrode to inject a relatively
 uniform current across the sample, and we measure the conductance with the local electrodes E5 and
 E6 (E7 and E8). **d**, Schematic illustration of our measurement geometry with current flow along x
 direction. We short the E5 and E6 as the drain electrode while E7 and E8 as the source electrode to
 inject a relatively uniform current across the sample, and we measure the conductance with the local
 electrodes E1 and E2 (E3 and E4). **e**, Conductance of channel 1 (E5 and E6) and channel 2 (E7 and
 E8) as a function of V_{tg-SiP_2} under the measurement geometry in (c). **f**, Conductance of channel 1
 (E1 and E2) and channel 2 (E3 and E4) as a function of SiP₂ top gate voltage under the measurement
 geometry in (d).

In short, we have addressed all the reviewer’s comments and suggestions point by point. We hope to
 convince the reviewer that our revised manuscript meets the criteria for publication in *Nature*
 *Communications*.

**Report of Reviewer #2 (Remarks to the Author)**

*In this manuscript, Li et al. reported a new non-symmetric dielectric material, SiP₂, for inducing*
*anisotropic electrical conductivity, linearly-polarized PL and anisotropic SHG in 2D MoS₂ that has*
*a symmetric C₃ structure. They attributed these phenomena to the anisotropic Moiré potential at the*
*SiP₂/TMDC interfaces. Such an idea of utilizing interface symmetry engineering on 2D*
*heterostructures and related field-effect transistor (FET) devices for emergent phenomena, which is*
*well-demonstrated in this work, is novel and inspiring for the study of condensed matter physics and*
*advanced electronic applications.*

*In general, this paper is well organized and well written. To the extent that I can appreciate, it is a*
*novel strategy to adopt such a non-symmetric gate dielectric material for inducing quite some*
*interesting emergent interfacial phenomena. For example, since the TMDC is a well-known material*
*with giant Berry curvature at K valleys, a modulation of the band structure of TMDC from the crystal*
*field of non-symmetric SiP₂ dielectric can cause the Berry curvature dipole and other effects*
*governed by the tunable interfacial symmetry. Thus, interesting interfacial physics and*
*corresponding phenomena can be expected by controlling and breaking the symmetry of the interface,*
*including the generation of the in-plane polarization (photovoltaic effect and quantum shift current),*
*the Berry curvature dipole (circular photo-galvanic effect and nonlinear anomalous Hall effect) and*
*the gauge field (Landau level quantization). Since those nonlinear properties usually only exist in*
*those low-symmetric interfaces, the report of such an anisotropic dielectric can provide a new*
*strategy for generating exotic functionalities at dielectric/semiconductor interfaces and related FET*
*devices via interfacial symmetry engineering.*

*In short, this is a fairly-good manuscript with high quality and broad interest, which would give a*
*substantial contribution to interface physics and twistrionics as well as advanced electronics. I would*
*like to recommend its publication in Nature Communications after clarifying several concerns below.*

**Authors' response:**

We appreciate the reviewer's positive evaluation and constructive comments on our manuscript,
which greatly helped us improve the quality of the manuscript. We have addressed all the comments
point by point in the revised manuscript, and thus we hope that the revised manuscript meets the
publication criteria of *Nature Communications*.

*1. The authors show the generation of giant anisotropy in SiP₂/TMDC interfaces by breaking the C₃*
*rotation symmetry of 1L-MoS₂ down to C₁ using a SiP₂ anisotropic dielectric with C₂ rotation*
*symmetry. This is very interesting. In principle, there can be in-plane polarization in such a C₁*
*interface (Science 372, 68-72 (2021)). The bulk photovoltaic effect is expected to occur at this*
*SiP₂/TMDC dielectric/semiconductor interface. This possibly arises from the shift current*
*mechanism due to the Berry connection at the interface. I suggest the authors discuss such possible*
*exciting phenomena in the prospect part of the manuscript.*

**Authors' response:**

We appreciate the reviewer's constructive suggestion and insightful comments. We totally agree with
the reviewer that there is in-plane polarization at our MoS₂/SiP₂ heterointerface with C₁ symmetry,
which is possible to generate a shift current due to the induced Berry connections at the MoS₂/SiP₂
interface and further lead to the bulk photovoltaic effect therein. The bulk photovoltaic effect in
solids is a kind of nonlinear optical process that converts light into electricity, which has a potential
advantage in solar cells with an efficiency that exceeds the fundamental Shockley–Queisser limit³.
This effect is only valid in crystals with broken inversion symmetry and can lead to significant
electronic polarization in photovoltaics. Unlike those polar materials^{4,5} that have intrinsically broken
inversion symmetry, the intentional breaking of inversion symmetry with symmetry-mismatched
heterostructures can provide a new effective way to induce giant in-plane polarization in materials
and thus lead to the bulk photovoltaic effect therein. Such an exciting phenomenon was recently
reported in the WSe₂/BP semiconductor/semiconductor system by symmetry engineering⁶. Actually,

the shift current (pointed out by the reviewer) and the anisotropic SHG results (which is studied in
this work) are identical from the symmetry view point and closely related with each other, since both
of them reflect the geometrical nature of the wave function⁷. Therefore, we believe the realization of
the bulk photovoltaic effect in our MoS₂/SiP₂ semiconductor/dielectric system would provide a
substantial contribution to interfacial physics and electronics, which as noticed by the reviewer, can
be a good research topic and is exactly what we are currently doing.

By following the reviewer's suggestion, we added several sentences in the prospect part (page 14)
of the revised manuscript to emphasize the novelty of this work and the interesting interfacial
phenomena (such as the bulk photovoltaic effect) that can be expected in this symmetry breaking
heterointerface: "*The tunable interfacial symmetry in the TMDC/SiP₂ heterostructure can provide a
unique platform for investigating symmetry-related interfacial physics and corresponding
phenomena, including the generation of the in-plane polarization (photovoltaic effect and quantum
shift current) and the Berry curvature dipole (circular photo-galvanic effect and nonlinear
anomalous Hall effect).*"

*2. Similarly, there is also a possibility of generating the Berry curvature dipole. Since the TMDC is
a well-known material with giant Berry curvature at K valleys, a slight modulation to the band
structure can cause the Berry dipole (Nature Physics 14, 900–906 (2018)). In addition to the
anisotropic conductance, one can also expect the nonlinear anomalous Hall effect at this SiP₂/TMDC
interface due to the symmetry breaking by the SiP₂ anisotropic dielectric. The circular photo-
galvanic effect is also expected, possibly emerging from the Berry curvature dipole at this
SiP₂/TMDC interface. Is there any possible experimental sign to confirm that?*

**Authors' response:**

We appreciate the reviewer for raising his/her professional comment on the possible generation of
Berry curvature dipole (BCD) in SiP₂/TMDC systems and resulting nonlinear effects. Actually, as
noticed by the reviewer, we indeed observed the experimental sign for a modulation to the band
structure caused by the Berry dipole and also confirmed the resulting nonlinear Hall effect and
circular photo-galvanic effect (CPGE) in our heterointerface.

Please allow us to explain it in detail.

As noticed by the reviewer, the Berry curvature dipole is a crucial topological quantity for
characterizing the electron wavefunction in solids and plays a significant role in realizing exotic
nonlinear phenomena of quantum materials. Owing to the imbalanced distribution of Berry curvature
on the band structure, the Berry curvature dipole directly determines one kind of second-order
response of quantum materials and can lead to the non-vanishing Hall conductivity even under the
preserved time-reversal symmetry⁸ [*Phys. Rev. Lett.* **115**, 216806 (2015)]. This pioneering work
inspired numerous high-level electronic transport studies on the nonlinear Hall conductivity of
different material systems^{9,10}, as exemplified by the Weyl semimetal of WTe₂ [*Nature* **565**, 337
(2019)] and twisted homo-bilayers of graphene [*Nat. Phys.* **18**, 765 (2022)]. In addition to the BCD-
induced nonlinear electronic transport phenomena, the Berry curvature dipole should also generate
an emergent nonlinear optical response (for example, the spin photocurrent) since the optical
transition probability is proportional to the value of Berry curvature during the photon-to-electron
angular-momentum transfer. In particular, in our symmetry-mismatched van der Waals
heterointerfaces, the band structure with interfacial hybridization of atomic orbitals is more sensitive
to Berry curvature dipole generation due to symmetry breaking. Therefore, BCD generation and the
induced spin photocurrent¹¹ or nonlinear Hall effect¹² at the SiP₂/TMDC heterointerface should be
expected.

By following the reviewer's suggestion, we performed nonlinear anomalous Hall measurements to

confirm that the BCD (one of the ongoing research topics, not the main focus of this study) can also
 be generated at our symmetry-mismatched heterointerfaces. In our new control experiments, we
 fabricated a Hall-bar device based on the MoS₂/SiP₂ heterointerface (*C*_{1v} interfacial symmetry).
 Based on the measurement geometry shown in Fig. R3a, we applied the alternating AC electrical
 current I_{ds}^{ω} perpendicular to the mirror plane of the heterointerface. In such a case, the BCD is
 perpendicular to the mirror plane and results in the emergence of a nonlinear anomalous Hall signal
 parallel to the mirror plane. As shown in Fig. R3b, the Li-electrolyte gated heterointerface shows
 good metallic behavior at $V_G = 2.4$ V, in which the Fermi level is tuned to be above the conduction
 band minimum. Note that the nonlinear anomalous Hall signal $V_{xy}^{2\omega}$ is probed by the second harmonic
 component via the lock-in amplifier (Fig. R3c) and found to be proportional to the square of the
 electrical current I_{ds}^{ω} ($V_{xy}^{2\omega} = bI_{ds}^{\omega 2}$ in which the nonlinear coefficient $b = 19.7$ k Ω/A^2). The linear
 transverse signal V_{xy}^{ω} is thus simultaneously measured for the Hall effect under a magnetic field with
 another lock-in amplifier, and the carrier density is estimated as $n_e = 1.7 \times 10^{14}/\text{cm}^2$ (Fig. R3d). Thus,
 the V_G -dependent carrier density n_e and the V_G -dependent nonlinear coefficient b are obtained (Fig.
 R3e,f). Such observations of the nonlinear Hall effect provide solid evidence that the BCD can be
 generated at the symmetry-mismatched heterointerface, as proposed by the reviewer.

 **Figure R3 | The nonlinear anomalous Hall effect at the MoS₂/SiP₂ heterointerface.** **a**, Schematic
 geometry for transport measurement, in which the AC current I_{ds}^{ω} is applied and the nonlinear
 anomalous Hall signal $V_{xy}^{2\omega}$ is probed by the second harmonic component. **b**, The temperature-
 dependent resistance of the Li-electrolyte gated heterointerface at $V_G = 2.4$ V. The inset is the optical
 microscopic image of the Hall-bar device of MoS₂/SiP₂ with *C*_{1v} symmetry, in which the region of
 monolayer MoS₂ is indicated by the blue dashed polygon. **c**, The I_{ds}^{ω} -dependent nonlinear anomalous
 Hall signal $V_{xy}^{2\omega}$. The solid curve is the parabolic fitting curve for $V_{xy}^{2\omega} = bI_{ds}^{\omega 2}$. **d**, The linear
 transverse signal V_{xy}^{ω} . **e,f**, The gate-dependent carrier density n_e and the gate-dependent nonlinear
 coefficient b . Such observations of the nonlinear Hall effect provide solid evidence that BCD can be
 generated at the symmetry-mismatched heterointerface.

By following the reviewer's suggestion, we performed the circular photo-galvanic current
 measurements based on the WSe₂/SiP heterostructure with symmetry breaking similar to that of our
 SiP₂/TMDC interface. And we indeed observe the circular photo-galvanic effect in such a symmetry-

mismatched heterointerface (This work is recently published in *Nature Nanotechnology*)¹³. Figure
 R4 shows our results of the spin photocurrents for a heterointerface under different configurations of
 incident light. The polarization-dependent photocurrent J can be quantitatively analyzed based on
 the following relation³⁰: $J = C \sin 2\varphi + L_1 \sin 4\varphi + L_2 \cos 4\varphi + D$, where C accounts for the spin
 photocurrent originating from the BCD, L_1 and L_2 account for the linear photocurrent corresponding
 to the shift current and thermal photocurrent²¹, and D accounts for a polarization-independent
 photocurrent. One can see that the circular photo-galvanic current (the component C) can be clearly
 observed in three different geometries. Such observations of the circular photo-galvanic effect
 provide experimental evidence that BCD can be generated at the symmetry-mismatched
 heterointerface, as proposed by the reviewer.

**In short, both the nonlinear Hall effect and CPGE effect emerging from the Berry curvature**
 **dipole are experimentally confirmed in our symmetry-mismatched heterointerfaces.** To address
 the reviewer's suggestion, we added several sentences in the prospect part (page 14) of the revised
 manuscript to discuss these possible interesting interfacial phenomena that can be expected in this
 symmetry-breaking interface: “*The tunable interfacial symmetry in the TMDC/SiP₂ heterostructure*
 *can provide a unique platform for investigating symmetry-related interfacial physics and*
 *corresponding phenomena, including the generation of the in-plane polarization (photovoltaic effect*
 *and quantum shift current) and the Berry curvature dipole (circular photo-galvanic effect and*
 *nonlinear anomalous Hall effect).”*

**Figure R4 | Geometric configuration of the spin photocurrent generated from the**
 **heterointerface. a–f,** Spin photocurrent J_x as a function of φ with $\lambda = 1064$ nm, $V_{ds} = 0$ V, and $V_G =$
 0 V. In the schematic figures of the experiment (top), the x direction is defined perpendicular to the
 mirror plane (highlighted in blue) of the interface. The direction of the incident light was set in the z
 direction (a–b), in the xz plane (c–d) and in the yz plane (e–f). The original data in each panel are
 shown with black open circles, and the total fitting function in each panel is shown with black lines.
 The spin photocurrent and linear photocurrent are shown with red and blue circles, respectively.
 Correspondingly, the fitted parameters of the spin photocurrent component C and linear photocurrent
 components (L_1 and L_2) are shown in the bar plot on the right side. The error bars are defined by the
 residual values of error analysis in Fourier series expansion for the original data.

*3. The band alignment is crucial for the heterostructure, which determines many physical processes,*
 *such as charge transfer, band bending effect, and so on. In Fig. 4d, the SiP₂/TMDC*
 *dielectric/semiconductor interface exhibits type-I band alignment. While in the supplementary, the*
 *author shows the type-II band alignment. What is the actual band alignment? I hope the author can*
 *experimentally identify the band alignment through Kelvin-probe force microscopy (KPFM) or*

**Authors' response:**

We appreciate the reviewer for raising his/her concern about the band alignment of 1L-MoS₂/SiP₂
heterostructure. Based on our DFT calculations and newly-added Kelvin-probe force microscopy
(KPFM) measurements, the actual band alignment is confirmed to be type-II band alignment.

Please allow us to explain it in detail.

Figure R5 shows the KPFM results of the 1L-MoS₂/SiP₂ heterostructure. The work functions of the
monolayer MoS₂ and SiP₂ flake are estimated to be approximately 4.61 eV and 4.96 eV. Based on
the fact that SiP₂ (1L-MoS₂) is a semiconductor with a bandgap of 2.25 eV (2.11 eV) according to
our calculations and the assumption that their Fermi level is sitting at the middle of the bandgap, we
confirm that the energy levels of the conduction band minimum and the valence band maximum of
SiP₂ are sitting at -3.48 eV and -5.73 eV (as indicated by the solid lines in Fig. R5e), while the
conduction band minimum and the valence band maximum of 1L-MoS₂ are sitting at -3.90 eV and
282 -6.01 eV (as indicated by the solid lines in Fig. R5e). Therefore, we can verify that the 1L-MoS₂/SiP₂
heterostructure forms a type-II band alignment.

In our DFT calculations with the HSE06 functional which correctly describes the band gaps, we
confirm that the energy levels of the conduction band minimum and the valence band maximum of
SiP₂ flake are sitting at -3.76 eV and -6.01 eV (as indicated by the solid lines in Fig. R5f), while the
conduction band minimum and the valence band maximum of 1L-MoS₂ are sitting at -4.14 eV and
288 -6.25 eV (as indicated by the solid lines in Fig. R5f). Herein, we set the energy position of vacuum
level as zero. If we compare the conduction and valence band offsets of these two materials, we can
verify that the 1L-MoS₂/SiP₂ heterostructure forms a type-II band alignment at the interface. Such a
result is well-consistent with our KPFM results.

Regarding the calculation result with the HSE06 functional showing type-I band alignment in Fig.
4d in the original manuscript, as we have explained in the Supplementary Information, this is
associated with the difference in conduction and valence band offset between the strained and
unstrained 1L-MoS₂/1L-SiP₂ interface in our simulated models. Due to the large lattice mismatch of
MoS₂ and SiP₂, the direct simulation of the large-size moiré lattice by using DFT is too expensive to
afford. Therefore, we constructed a heterostructure with strained MoS₂ and SiP₂ (Fig. 4d) to simulate
the influence of the moiré potential on the electronic states of 1L-MoS₂ in an affordable way. Such
a strain will cause the change in the valence band offset between 1L-MoS₂ and 1L-SiP₂ artificially
and make the band alignment change from type-II to type-I. Note that even though such a valence
band offset is reversed by applied artificial strain and the band alignment changes to type-I, the
conduction band offset is not influenced and the conduction band edge is still contributed by the
MoS₂ layer. Such a preserved conduction band offset enables us to investigate the transport behavior
of the electrons correctly with the strained slab models.

In our previous manuscript, we put the strained 1L-MoS₂/1L-SiP₂ heterostructure with type-I band
alignment in both Fig. 4d of the main text and in the Supplementary Information while presenting
the unstrained 1L-MoS₂/1L-SiP₂ heterostructure with type-II band alignment in the Supplementary
Information.

To avoid any misunderstanding, we have revised the Fig. 4d and the related captions in the revised
manuscript. To emphasize that the real band alignment of the 1L-MoS₂/SiP₂ heterostructure is type-
II, we added the KPFM results as Fig. S22 and the corresponding discussion on page 38 of the revised
Supplementary Information:

*“To experimentally determine the band alignment of the 1L-MoS₂/SiP₂ heterostructure, we performed*
*additional Kelvin probe force microscopy (KPFM) measurements. One can see in Supplementary*
*Fig. 22c that the work functions of the monolayer MoS₂ and SiP₂ flake are estimated to be*
*approximately 4.61 eV and 4.96 eV, respectively. Based on the fact that SiP₂ (1L-MoS₂) is a*
*semiconductor with a band gap of 2.25 eV (2.11 eV) according to our calculations and the*
*assumption that their Fermi level is sitting at the middle of the band gap, we confirm that the energy*

levels of the conduction band minimum and the valence band maximum of SiP₂ are sitting at -3.48
 320 eV and -5.73 eV, respectively (as indicated by the solid lines in Supplementary Fig. 22f), while the
 321 conduction band minimum and the valence band maximum of 1L-MoS₂ are sitting at -3.90 eV and
 322 -6.01 eV, respectively (as indicated by the solid lines in Supplementary Fig. 22f). Therefore, we can
 verify that the 1L-MoS₂/SiP₂ heterostructure forms a type-II band alignment. Such a result is well-
 consistent with our DFT calculated results.”

**Figure R5 | Confirmation of the work functions and band alignment of the 1L-MoS₂/SiP₂**
 **heterostructure based on KPFM measurements.** **a**, Optical microscopy image of the 1L-
 MoS₂/SiP₂ heterostructure on Au film. The regions of monolayer 1L-MoS₂ and SiP₂ are highlighted
 by blue and white dashed polygons. The red solid square indicates the region for the AFM and KPFM
 measurements. **b**, Atomic force microscope image of the 1L-MoS₂/SiP₂ heterostructure. The regions
 of the monolayer MoS₂, Au film, 1L-MoS₂/SiP₂ heterostructure and SiP₂ flake are labeled. **c**, KPFM
 imaging of the 1L-MoS₂/SiP₂ heterostructure. The red and blue lines indicate the profiles from 1L-
 MoS₂ to Au film and from SiP₂ to Au film. **d**, The profiles from 1L-MoS₂ to Au film (red) and from
 SiP₂ to Au film (blue) in the KPFM measurement. **e**, Schematic figure for type-II band alignment
 between monolayer MoS₂ and SiP₂, confirmed from the above KPFM measurements. The dashed
 lines indicate the energy levels of the Fermi level. The experimental estimations for the conduction
 band minima and the valence band maxima of 1L-MoS₂ and SiP₂ are labeled. **f**, Schematic figure for
 type-II band alignment between monolayer MoS₂ and SiP₂ according to the energy levels obtained
 from our first-principles calculations.

*4. The authors demonstrate a giant anisotropic conductance in the SiP₂-gated MoS₂ FETs due to the*
 *formation of anisotropic moiré potential at SiP₂/MoS₂ interface via symmetry engineering, which is*
 *very interesting. I wonder what is the difference between the results of temperature-dependent*
 *conductance/resistance along the two different directions of the SiP₂-gated MoS₂ FETs?*

**Authors' response:**

We appreciate the reviewer for raising his/her comment on the temperature-dependent resistance
 along two directions of SiP₂-gated MoS₂ FETs. Due to the formation of anisotropic moiré potential
 at MoS₂/SiP₂ heterointerfaces, the temperature-dependent resistance along two different directions
 of SiP₂-gated 1L-MoS₂ FETs can be remarkably different. For example, in the gating-induced

insulator-to-metal transition of the heterointerface, we find a typical hysteresis-free temperature-
 dependent resistance along x direction while confirming a possible “hidden order” with large
 hysteresis in the temperature-dependent resistance along y direction.

Please allow us to explain it in detail.

To investigate the SiP₂-gating behavior along two different directions of SiP₂/MoS₂ heterostructure,
 we measured the temperature-dependent resistance with current flowing along x and y directions. As
 shown in Fig. R6a, when the current is applied along the x direction, the warming-up and cooling-
 down resistances of SiP₂-gated 1L-MoS₂ under each $V_{\text{tg-SiP}_2}$ are almost the same without any
 hysteresis, and one can see a typical insulator-to-metal transition with increased $V_{\text{tg-SiP}_2}$.
 Interestingly, in the geometry with current flowing along y direction, the warming-up and cooling-
 down resistance of SiP₂-gated 1L-MoS₂ can be remarkably different with a large hysteresis. Such
 hysteresis in temperature-dependent resistance is similar to those in charge density wave materials¹⁴,
 such as 1T-TaS₂. Hence, we speculate that there might be a “hidden order” induced by the anisotropic
 moiré potential along y direction of SiP₂/1L-MoS₂. Note that similar current direction-related
 resistance behavior is also observed in the EuO/KTO (111) interface¹⁵, which is attributed to the
 formation of a unique stripe phase along the $[1\ 1\ \bar{2}]$ of the oxide interface. In contrast, for those SiP₂-
 gated devices based on 20-nm MoS₂ (Fig. R6c,d), there is no observation of such “hidden order”
 behavior since the sample is too thick, which indicates that this is indeed an emergent interfacial
 phenomenon. The observation of “hidden order” with large hysteresis in the temperature-dependent
 resistance suggests that there could be abundant correlated physics in such symmetry-mismatched
 heterointerfaces, which can be a good research topic and will be systematically discussed in our
 future paper.

 **Figure R6 | Temperature-dependent resistance of the MoS₂/SiP₂ heterostructure under various**
 **SiP₂ top gate voltages with current flow along the x and y directions of the heterostructure. a,**
 **b,** Temperature-dependent resistance of the 1L-MoS₂/SiP₂ heterostructure under various SiP₂ top
 gate voltages with current flow along the x (a) and y (b) directions of the heterostructure. The inset
 is the optical image of the 1L-MoS₂/SiP₂ device. **c,d,** Temperature-dependent resistance of the 20-
 377 nm-MoS₂/SiP₂ heterostructure under various SiP₂ top gate voltages with current flow along the x (c)
 and y (d) directions of the heterostructure. The inset is the optical image of the 20-nm-MoS₂/SiP₂
 device.

*5. The trion states in 1L-TMDC are widely-observed due to their large binding energies and always*
 *provide an ideal platform to study excitonic many-body physics. The authors show that the exciton*
 *in 1L-TMDC becomes linearly-polarized in the SiP₂/TMDC dielectric/semiconductor interface via*
 *symmetry engineering. How about trion states in this system? Does the trion state also show such*
 *linear polarization due to symmetry breaking?*

**Authors’ response:**

We appreciate the reviewer for raising his/her comment about the trion state in the SiP₂/TMDC
 dielectric/semiconductor heterostructure system. Yes, the trion state also shows linear polarization at
 the SiP₂/TMDC heterostructure. As shown in the polarization-resolved PL measurements in Fig. R7,
 both the exciton and trion states of 1L-WS₂ are linearly polarized along the *x* direction of the
 heterostructure. The degree of linear polarization *P* in 1L-WS₂, which is defined as $P = (I_{\max} -$
 $I_{\min}) / (I_{\max} + I_{\min})$, is ~ 0.16 for excitons and ~ 0.13 for trions, where I_{\max} (or I_{\min}) is the maximum (or
 minimum) PL emission intensity. The dramatic contrast in the polarized PL emission of trion states
 (polarized state with SiP₂ dielectric and non-polarized state without SiP₂ dielectric) indicates that
 such symmetry engineering can be extended even to those bound excitonic states.

**Figure R7 | Observation of anisotropic trion state in 1L-WS₂/SiP₂.** **a**, PL spectra of 1L-WS₂/SiP₂
 with an analyzer along the *x* direction. The blue, green, and red shadows represent the contributions
 of the exciton (*X*), trion (*T*), and trapped states in 1L-WS₂. Inset: optical image of the 1L-WS₂/SiP₂
 heterostructure. The red (or yellow) dashed line represents the WS₂ (or SiP₂) sample. Scale bar is 10
 399 μm. **b**, Polarization-resolved PL integrated intensities of the exciton of 1L-WS₂ on SiO₂.
 Polarization-resolved PL integrated intensities of excitons (**c**) and trions (**d**) in 1L-WS₂/SiP₂. The
 solid circle is the experimental data, and the solid line is a fitting result using a $\cos^2\theta$ function.

To address the reviewer's comment on the trion state in the SiP₂/TMDC heterostructure, we added
 the polarization-resolved PL result (Fig. S8) and corresponding discussions on page 18 of the revised
 Supplementary Information:

*"To investigate whether the trion state in TMDCs can also be modulated by the anisotropic SiP₂*
 *dielectric when forming the SiP₂/TMDC heterostructure, we performed polarization-resolved PL*
 *measurements in another 1L-WS₂/SiP₂ heterostructure. As shown in Supplementary Fig. 8a, both the*
 *exciton and trion states of 1L-WS₂ are linearly polarized along the *x* direction of the heterostructure.*
 *The degree of linear polarization *P* in 1L-WS₂ is ~ 0.16 for excitons and ~ 0.13 for trions. The*

dramatic contrast in the polarized PL emission of trion states (polarized state with SiP₂ dielectric
 and non-polarized state without SiP₂ dielectric) indicates that such symmetry engineering can be
 extended even to those bound excitonic states.”

Also, I suggest the authors consider addressing the following technical issues:

1. In Fig. 1c and d, the authors show the transfer curves and leakage current of the 5-nm-thick MoS₂
 FETs. How about the results of the 1L-thick device? And Fig. 1f shows the R_{xx}-T curves tuned by
 adjusting the SiP₂ gate for the 1L-MoS₂ FETs. How about the results for thicker samples?

**Authors’ response:**

We appreciate the reviewer for raising his/her concern about thickness-dependent transport behavior
 of SiP₂-gated MoS₂ devices.

Regarding the concern on the *transfer curves and leakage current of the 1L-MoS₂ FETs*, we measured
 the transfer characteristics of 1L-MoS₂ transistors with a SiP₂ top gate. As shown in Fig. R8a, when
 sweeping the top gate voltage V_{tg-SiP₂} up to 4 V, the 1L-MoS₂ transistor shows an on/off ratio as
 high as 10⁵ (similar to the 5-nm-MoS₂). The measured leakage current density of the SiP₂-gated 1L-
 MoS₂ transistor is as small as approximately 10⁻⁵ A cm⁻² at an external electric field strength of 1.2
 426 MV cm⁻¹ (Fig. R8b). Such a low leakage current is also comparable to those of transistors gated by
 427 high-κ dielectrics¹⁶⁻¹⁸ such as Al₂O₃, HfO₂ or Bi₂SeO₅ and better than the criteria of the low-power
 limit and the standard complementary metal–oxide–semiconductor gate limit¹⁸.

**Figure R8 | High-performance MoS₂ transistors gated with a SiP₂ dielectric with non-**
 **symmorphic C₂ rotational symmetry.** **a**, Transfer curves of the 1L-MoS₂-based transistor at 2 K
 when sweeping top gate voltages via the SiP₂ dielectric. **b**, Leakage current I_{gs} as a function of
 V_{tg-SiP₂} on a 1L-MoS₂ device at 2 K. I_{gs} and V_{tg-SiP₂} are rescaled to the leakage current density and
 the electric field strength for a better comparison (right and top axes). Horizontal red lines mark the
 limits of leakage current density for various types of integrated circuits. **c**, Four terminal resistances
 (R_{xx}) versus temperature of a 20-nm-MoS₂ device under different V_{tg-SiP₂} values.

Regarding the concern on the *SiP₂-gated $R_{xx}-T$ of the thicker MoS₂ FETs*, we measured the
temperature-dependent four-terminal resistance ($R_{xx}-T$) of a 20-nm-MoS₂/SiP₂ transistor. As shown
in Fig. R8c, the $R_{xx}-T$ curves show typical insulating behavior with negative temperature
coefficients dR_{xx}/dT when $V_{tg-SiP_2} < 3.8$ V. In contrast, when $V_{tg-SiP_2} > 3.8$ V, R_{xx} starts to
decrease with cooling temperature from 150 K to 100 K, and the positive dR_{xx}/dT shows the typical
metallic behavior in SiP₂-gated 20-nm-MoS₂. Such a gate-tunable insulator-metal transition in 20-
443 nm-MoS₂ demonstrates the great dielectric capability of SiP₂ in MoS₂ with different thicknesses.

To address the reviewer's concern on the transport results of devices with different MoS₂ thicknesses,
we added the related results in Fig. S2 and corresponding discussions on page 6 of the revised
Supplementary Information:

*“Supplementary Fig. 2a shows the transfer characteristics of 1L-MoS₂ transistors with a SiP₂ top*
*gate. With sweeping V_{tg-SiP_2} up to 4 V, the 1L-MoS₂ transistor shows an on/off ratio as high as 10^5*
*(similar to the 5-nm-MoS₂). The measured leakage current density of the SiP₂-gated 1L-MoS₂*
*transistor is as small as approximately 10^{-5} A cm^{-2} at an external electric field strength of 1.2 MV*
*cm^{-1} (Supplementary Fig. 2b).”*

*2. In Fig. 1d, the authors label the right axis as “current” for the leakage current of the SiP₂-gated*
*MoS₂ FETs. However, the unit of right axis is “A cm^{-2} ”. Do the authors want to show the “current*
*density” here? The authors should redefine the label of the right axis precisely.*

**Authors' response:**

We appreciate the reviewer's comment on the label of the right axis of Fig. 1d. As noticed by the
reviewer, we do want to show the leakage current density in Fig. 1d since it is widely used to evaluate
device performance. By following the reviewer's suggestion, we redefined the label of the right axis
of Fig. 1d precisely as the “leakage current density J_{Leak} ” in the revised manuscript.

*3. In Fig. 1f, the authors show the insulator-metal transition in SiP₂-gated MoS₂ FETs. One missing*
*piece of information is the sheet resistivity, R_c (the authors only show four-terminal resistance R_{xx} ,*
*which is not universal, while R_c should be) that can be measured accurately — all we need are the*
*sample dimensions — and such a value is much more informative than the resistance R_{xx} .*

**Authors' response:**

We appreciate the reviewer for raising this important issue. By following the reviewer's suggestion
about using the sheet resistivity, R_s , we replace the resistance R_{xx} with the sheet resistivity R_s and
replot the data in Fig. 1f in the revised manuscript.

*4. In Fig. 2c, the authors show that the isotropic SHG signal in 1L-MoS₂ becomes anisotropic in*
*SiP₂/1L-MoS₂ heterointerface. Does the thickness of SiP₂ influence this phenomenon, such as the*
*anisotropic ratio of the SHG signal? Can the authors discuss the thickness effect of SiP₂ on such*
*anisotropic SHG signals?*

**Authors' response:**

We appreciate the reviewer for raising his/her concern about the influence of SiP₂ thickness on the
anisotropic ratio of the SHG signal in the 1L-MoS₂/SiP₂ heterostructure. A short answer is that the
anisotropic SHG response in the heterostructure is robust with the change of the SiP₂ thickness and
the anisotropic ratio shows little sensitivity to the SiP₂ thickness.

Please allow us to explain it in detail.

To investigate the influence of SiP₂ thickness on the anisotropic SHG response on the 1L-MoS₂/SiP₂
heterostructure, we performed polarized SHG measurements on 1L-MoS₂/SiP₂ by changing the
bottom SiP₂ thickness. As shown in Fig. R9, the anisotropic SHG response exists in all the samples
with SiP₂ thicknesses ranging from 10 nm to 30 nm. More importantly, the anisotropy in the SHG

signal changes little with increasing SiP₂ thickness. Such a robust anisotropic SHG response in 1L-
MoS₂/SiP₂ and its independence of SiP₂ thickness can be expected since such a phenomenon should
only be determined by symmetry breaking at the 1L-MoS₂/SiP₂ heterointerface between the 1L-
MoS₂ and the topmost SiP₂ layer and should not be influenced by the thickness of the SiP₂ substrate.

To address the reviewer's concern, we added related SHG results (Fig. S10) and the corresponding
discussion on page 20 of the revised Supplementary Information to show the SiP₂ thickness influence
on the anisotropic SHG response:

*“To investigate the influence of SiP₂ thickness on the anisotropic SHG response on the*
*heterostructure, we performed polarized SHG measurements on 1L-MoS₂/SiP₂ by changing the*
*bottom SiP₂ thickness. As shown in Supplementary Fig. 10, the anisotropic SHG response exists in*
*all the samples with SiP₂ thicknesses ranging from 10 nm to 30 nm. More importantly the anisotropy*
*in the SHG signal changes little with increasing SiP₂ thickness. Such a robust anisotropic SHG*
*response in 1L-MoS₂/SiP₂ and its independence of SiP₂ thickness can be expected since such a*
*phenomenon should only be determined by symmetry breaking at the 1L-MoS₂/SiP₂ heterointerface*
*between the 1L-MoS₂ and the topmost SiP₂ layer and should not be influenced by the thickness of the*
*SiP₂ substrate.”*

We also added the corresponding discussion sentences on page 8 of the revised manuscript: *“More*
*importantly, the anisotropic SHG response at 1L-MoS₂/SiP₂ changes little with increasing SiP₂*
*thickness (Supplementary Fig. 10), further confirming that this is an interfacial phenomenon induced*
*by symmetry breaking at the 1L-MoS₂/SiP₂ heterointerface between the 1L-MoS₂ and the topmost*
*SiP₂ layer.”*

**Figure R9 | Anisotropic SHG responses in 1L-MoS₂/SiP₂ with various SiP₂ thicknesses. a–c,**
**Polar plot of polarization-resolved SHG intensities of 1L-MoS₂/SiP₂ under the parallel configuration**
**(the detection polarization is parallel to the excitation polarization) with SiP₂ thicknesses of 10 nm**
**(a), 25 nm (b) and 30 nm (c). Top inset: optical image of the 1L-MoS₂/SiP₂ heterostructure. The red**
**cross highlights the 1L-MoS₂/SiP₂ region.**

*5. In Fig. 2d, the authors show that the exciton in 1L-MoS₂ becomes linearly-polarized in SiP₂/MoS₂.*
*However, one can see a lower energy peak on the side of the exciton peak in the PL spectra of the*
*SiP₂/1L-MoS₂ heterostructure, which also shows linear polarization. What is this lower energy peak*
*in SiP₂/MoS₂? Is it the interlayer exciton peak?*

**Authors' response:**

We appreciate the reviewer for raising his/her concern on the lower energy peak in the 1L-MoS₂/SiP₂
heterostructure. This lower energy peak in the 1L-MoS₂/SiP₂ heterostructure might be from the
defect states of 1L-MoS₂ and is not the interlayer exciton state. In order to prove this, we performed
power-dependent PL measurements on 1L-MoS₂/SiP₂ heterostructure (Fig. R10a,b) and noticed that
the lower energy peak at 1.78 eV can be attributed to the defect states associated with S vacancies¹⁹⁻
²². Two important features in the power-dependent PL spectra should be addressed here: 1) the PL

intensity of the exciton state at 1.91 eV monotonically increases with increasing excitation power, as
 shown in Fig. R10c, and 2) the PL intensity ratio between the exciton at 1.91 eV and the lower energy
 peak at 1.78 eV gradually increases with increasing excitation power (Fig. R10c). Such behavior is
 similar to those of pristine 1L-MoS₂ samples in previous reports²³ (Fig. R10d), indicating that the
 lower energy peak is related to the defect states from 1L-MoS₂.

To clarify the origin of the lower energy peak, we revised the corresponding discussion sentences on
 page 15 of the revised Supplementary Information: “When 1L-MoS₂ is placed on SiP₂, we find a
 strong lower energy peak on the side of the exciton peak in the PL spectra of the 1L-MoS₂/SiP₂. This
 exciton state might be related to the defect states in intrinsic 1L-MoS₂ and might be enhanced at the
 heterointerface.”

 **Figure R10 | Trapped exciton states at the 1L-MoS₂/SiP₂ heterointerface.** a, PL spectra of SiP₂
 (green curve, bottom panel), 1L-MoS₂ (blue curve, middle panel), and the 1L-MoS₂/SiP₂
 heterostructure (red curve, top panel) at 77 K. The defect exciton of 1L-MoS₂ is highlighted with an
 orange dashed line and a green arrow. b, Power-dependent PL spectra of the 1L-MoS₂/SiP₂
 heterostructure at 77 K. The PL spectra is normalized by the peak at 1.78 eV. c, Top panel: PL
 integrated intensity of exciton peak at 1.91 eV as a function of power of incident light. Bottom panel:
 PL integrated intensity ratio between the exciton peak at 1.91 eV and the peak at 1.78 eV as a function
 of power of incident light. d, Power-dependent PL spectra of a 1L-MoS₂ sample at 77 K. Figure is
 adopted from Ref. ²³.

6. Fig. 2e is the corresponding color plot of the PL intensity in Fig. 2d. In Fig. 2d, the PL intensity
 of SiP₂ (black curve) and 1L-MoS₂/SiP₂ heterostructure (red curve) appear to be comparable, but in
 Fig. 2e, why is there a five-fold difference in the intensity color scale between the two?

**Authors' response:**

There might be misunderstanding on the intensity color scale in Fig. 2d and Fig. 2e in the original
 manuscript (here Fig. R11a and Fig. R11b). When saying *In Fig. 2d, the PL intensity of SiP₂ and 1L-
 MoS₂/SiP₂ heterostructure appear to be comparable*, the reviewer might compare the PL intensity of
 defect exciton peak in SiP₂ (ranging from 1.3 eV to 1.6 eV, 440 counts) and exciton peak in 1L-
 MoS₂/SiP₂ at 1.90 eV (430 counts). However, the *five-fold difference in the intensity color scale* in
 the corresponding PL mapping data (Fig. R11b) is a comparison of the PL intensity of the intrinsic
 exciton peak in SiP₂ at 2.06 eV and the PL intensity of the exciton peak in 1L-MoS₂/SiP₂ at 1.90 eV.
 Actually, the PL intensity of the exciton peak in 1L-MoS₂/SiP₂ at 1.90 eV is about 430 counts, while
 the PL intensity of the intrinsic exciton peak in SiP₂ at 2.06 eV is about 110 counts (Fig. R11a). There
 is indeed a four- to five-fold difference in the PL intensity in the PL spectra, which is consistent with
 the PL mapping data in Fig. R11b.

**Figure R11 | Anisotropic PL response at the 1L-MoS₂/SiP₂ heterointerface.** **a**, PL spectra of SiP₂
 (green curve), 1L-MoS₂ (blue curve), and the 1L-MoS₂/SiP₂ heterostructure (red curve) at 77 K. The
 exciton emission of 1L-MoS₂ is highlighted with shadows. The PL mapping energy range is
 highlighted by the orange dashed rectangle. **b**, Corresponding color plot of the PL intensity in (a)
 as a function of emission photon energy at different detection polarization angles θ . Here, θ denotes the
 angle between the analyzer polarization direction and the x direction.

*7. In Fig. 4a, the Moiré period contains 12 hexagons in the y direction, while in Fig. 4g, it contains*
 *14 hexagons. What is the Moiré period (and the size of Moiré Brillouin zone) of this MoS₂/SiP₂*
 *interface?*

**Authors' response:**

We appreciate the reviewer for the concern on the moiré period (and the size of the moiré Brillouin
 zone) of this MoS₂/SiP₂ interface. In the experiments, the lattice constant of 1L MoS₂ is 3.16 Å in
 hexagonal lattice, which is corresponding to 3.16 Å in the x direction (a) and 3.47 Å in the y direction
 (b) in the $\sqrt{3} \times 1$ centered rectangular supercell. The lattice constant for 1L SiP₂ is $a = 3.44$ Å in the
 x direction and $b = 10.11$ Å in the y direction. Once we put 1L MoS₂ on the top of SiP₂, there is
 always a lattice mismatch and it is impossible to form the commensurate heterostructure in our model.
 In order to build a commensurate heterostructure model and show the atomic moiré pattern clearly,
 we need to artificially change the lattices of 1L MoS₂ and 1L SiP₂ slightly, but do not change their
 electronic structures too much. Specifically, for 1L-MoS₂, in the $\sqrt{3} \times 1$ centered rectangular
 supercell, a compressive strain (-0.008%) is applied along a , and the lattice constant b is enlarged
 from 3.47 Å to 3.48 Å. While for 1L-SiP₂, a tensile strain (0.2%) is introduced to a , and a compressive
 strain (-0.6%) is applied to b . Finally, the 12×11 MoS₂ centered rectangular lattice and 11×6
 SiP₂ lattice serve as the bricks to form the heterostructure models, where the lattice constants a and
 b in the commensurate heterostructure models are 37.917 Å and 60.282 Å, respectively. The
 magnitude of the corresponding reciprocal lattice vector \mathbf{a}^* is 0.166 \AA^{-1} , while that of the
 corresponding reciprocal lattice vector \mathbf{b}^* is 0.104 \AA^{-1} , which defines the size of the moiré Brillouin
 zone. We would like to mention that this value should be close to the experimental value but will not
 be exactly same, because there is no in-plane strain or stress to apply to the sample intentionally in
 the experiments.

On the other hand, even this commensurate heterostructure model, which contains 1617 atoms
 including the 12×11 MoS₂ centered rectangular lattices and 11×6 SiP₂ lattices, is too large to

simulate directly by using current DFT code. In order to simulate the stacking configurations in the
moiré patterns and their electronic structures directly, we need to build suitable heterostructure
models. These models should be large enough to contain several stacking configurations observed in
moiré patterns, such as AA, AB and BA stackings. On the other hand, these models should not be
too large and will be affordable for our DFT simulations. After carefully weighing these options and
restrictions, we artificially built four model structures, labeled as case-I-ABBA (Supplementary Fig.
24d), case-I-AA (Supplementary Fig. 24f), case-II-AABA (Supplementary Fig. 24e), and case-II-AB
(Supplementary Fig. 24g). As shown in Supplementary Fig. 24a and b, a 14×1 MoS₂ ($\sqrt{3} \times 1$
centered rectangular supercell) and 13×1 SiP₂ are used to construct the heterostructure models, in
which the lattice constants a and b are 10.69 Å and 45.5 Å, respectively. In these models, such as
case-I-ABBA (Fig. 4g in the original manuscript mentioned by the reviewer), the electronic
structures for AB and BA stackings that exist in the moiré pattern of case-I (Fig. 4a) can be simulated
directly. Thus, we cannot use the lattice in these artificial models to evaluate the size of the moiré
period directly.

To address the reviewer's comment about the size of the moiré period, we revised the corresponding
sentences on page 35 of the revised Supplementary Information:

*“Due to the large lattice mismatch, we apply a suitable strain and stress to 1L-MoS₂ and 1L-SiP₂,
obtaining the commensurate 1L-MoS₂/1L-SiP₂ heterostructure models to show the moiré pattern.
Specifically, for 1L-MoS₂ with $\sqrt{3} \times 1$ centered rectangular supercell, a compressive strain (–
0.008%) is applied along the lattice constant a , and the lattice constant b is enlarged from 3.47 Å to
3.48 Å. While for 1L-SiP₂, a tensile strain (0.2%) is introduced to the lattice constant a , and a
compressive strain (–0.6%) is applied to the lattice constant b . Finally, the 12×11 MoS₂ centered
rectangular lattice and 11×6 SiP₂ lattice serve as the bricks to form the heterostructure models,
where the lattice constants a and b of the commensurate heterostructure models are 37.917 Å and
60.282 Å, respectively. The magnitude of the corresponding reciprocal lattice vector \mathbf{a}^* is 0.166 Å^{-1} ,
while that of the corresponding reciprocal lattice vector \mathbf{b}^* is 0.104 Å^{-1} , which defines the size of the
moiré Brillouin zone.”.*

In short, we have addressed all the reviewer's comments and suggestions point by point. We hope to
convince the reviewer that our revised manuscript meets the criteria for publication in *Nature*
*Communications*.

**Report of Reviewer #3 (Remarks to the Author)**

*In this manuscript by Z. Li et al., the authors report the realization of anisotropic optical and*
*transport properties in 2D materials by engineering the C_3 rotational symmetry breaking at*
*heterointerfaces. The authors fabricated $\text{SiP}_2/\text{MoS}_2$ heterostructure devices, where the SiP_2 layer*
*serves not only as a dielectric gate medium, but also as a C_3 symmetry-breaking factor. Due to the*
*proximal SiP_2 characterized by non-symmorphic C_2 symmetry, the intrinsically isotropic MoS_2 layer*
*evinced linearly-polarized photoluminescence, anisotropic second-harmonic generation, and*
*anisotropic conductance. With additional DFT calculations that suggest the MoS_2 lattice*
*deformation and charge density distribution having mirror symmetry, the authors concluded that the*
*formation of moiré potentials at the $\text{SiP}_2/\text{MoS}_2$ interface results in the observed anisotropies.*

*Engineering rotational symmetry breaking in 2D materials and their heterostructures define a*
*versatile strategy to create emerging optical, electrical, magnetic, and topological properties. Recent*
*examples include engineering uniaxial strain to control the optical responses and magnetism of 2D*
*materials, and forming moiré wires or rectangles to achieve highly-anisotropic transport and optical*
*responses in twisted bilayer systems; this current manuscript by Z. Li et al. made an exciting and*
*unique addition to the progress on engineering rotational symmetry breaking of 2D materials. The*
*C_3 rotational symmetry of 2D materials was broken by a dielectric medium with C_2 symmetry.*

**Authors' response:**

We appreciate the reviewer's professional evaluation and positive comments on our manuscript,
which greatly helped us improve the quality of the manuscript. We have addressed all the comments
point by point in the revised manuscript, and thus we hope that the revised manuscript meets the
publication criteria of *Nature Communications*.

*While the authors have demonstrated the resultant, highly-anisotropic optical and transport*
*properties in $\text{SiP}_2/\text{MoS}_2$, the discussion regarding the fundamental origin of such observations*
*requires further experimental evidence. The authors attributed the origin of the anisotropic*
*conductance index and linearly-polarized optical responses to the formation of moiré superlattices.*
*This argument is inferred from DFT calculations, which suggest the variation of interlayer distances*
*and the charge density distributions having a mirror symmetry due to the modulation by moiré*
*superlattices. The authors argued that the conduction-band-edge charge density distributions*
*imprinted by the moiré result in anisotropic charge (e.g., electrons doped at the band edge) hopping,*
*and the exciton PL polarization. However, no direct experimental evidence supports this argument,*
*e.g., the morphology of the moiré lattice and how the anisotropic properties vary as a function of*
*twist angles.*

**Authors' response:**

We appreciate the reviewer for raising this concern. By following the reviewer's suggestions, we
performed new optical experiments to show how the anisotropic properties of the heterostructure
vary as a function of twist angle. We succeed in confirming that the anisotropic PL and SHG
responses indeed strongly depend on the twist angle between MoS_2 and SiP_2 , and the corresponding
anisotropy dramatically decreases when the mirror symmetry of moiré superlattice is broken with
the change of the twist angle from 0° to 30° . These new experimental evidence supports our
theoretical argument based on the DFT calculations to a large extent.

Specifically, in those 1L- $\text{MoS}_2/\text{SiP}_2$ heterostructures with different twist angles (for example 0° and
50° shown in Fig. R12), the polarized SHG signals display strong anisotropic behavior, which is
remarkably different from the six-fold symmetric SHG signals in pristine 1L- MoS_2 . Such contrasting
behavior before and after stacking SiP_2 to 1L- MoS_2 indicates that the C_3 symmetry of 1L- MoS_2 is
broken by the C_2 symmetry of the bottom SiP_2 . More importantly, if we compare the polarized SHG
signals between those cases with mirror symmetry and without mirror symmetry, one can see that
the 0° case with mirror symmetry shows stronger anisotropy in SHG signals than that of the 50° case
without mirror symmetry. Such a twist-angle-dependent anisotropic SHG response demonstrates that

the mirror symmetry at MoS₂/SiP₂ heterointerface plays a key role in generating anisotropic optical
responses at the moiré superlattice.

As shown in the polarization-resolved PL results for 1L-MoS₂/SiP₂ heterostructures with various
twist angles (Fig. 13a-f), two important features can be observed as follows: 1) the anisotropic PL
responses exist at all twist angles; 2) the polarization degree ($P = \frac{I_{\max} - I_{\min}}{I_{\max} + I_{\min}}$, where I_{\max} and I_{\min} are
the maximum and minimum PL intensity) of the PL, which can qualitatively reflect the anisotropic
index of PL, shows a strong dependence on the twist angle. For example, the polarization degree P
is 0.33 for the 0° case and decreases to 0.14 as the twist angle increases up to 30°. Such a decrease
in the anisotropic index of PL might be related to the vanishing of the mirror symmetry with twist
angle changing from 0° to 30°. This result indicates that the mirror symmetry of MoS₂/SiP₂ moiré
superlattice plays an important role in controlling the magnitude of anisotropic optical responses at
the heterointerface.

**Figure R12 | Twist-angle-dependent anisotropic SHG response at 1L-MoS₂/SiP₂**
**heterostructure. a,b** Polar plot of polarization-resolved SHG intensities of 1L-MoS₂/SiP₂ under the
parallel configuration (the detection polarization is parallel to the excitation polarization) with a twist
angle of 0° (a) and 50° (b). The solid lines represent the fitting and the balls represent the
experimental data.

To address reviewer's suggestion on twist-angle-dependent properties, we added the twist-angle-
dependent SHG and PL signals (Figs. S11 and S12) and corresponding discussion on page 22 of the
revised Supplementary Information:

*“To investigate the effect of twist angle on the anisotropic properties of the heterostructure, we*
*performed polarization-resolved PL and SHG measurements on various 1L-MoS₂/SiP₂*
*heterostructures with different twist angles. Specifically, in those 1L-MoS₂/SiP₂ heterostructures with*
*different twist angles (for example 0° and 50° shown in Supplementary Fig. 11), the polarized SHG*
*signals displays the strong anisotropic SHG behavior, which is remarkably different from the six-fold*
*symmetric SHG signals in pristine 1L-MoS₂. Such contrasting behavior before and after stacking*
*SiP₂ to 1L-MoS₂ indicates that the C₃ symmetry of 1L-MoS₂ is broken by the C₂ symmetry of the*
*bottom SiP₂. More importantly, if we compare the polarized SHG signals between those cases with*
*mirror symmetry and without mirror symmetry, one can see that the 0° case with mirror symmetry*
*shows stronger anisotropy in SHG signals than that of the 50° case without mirror symmetry. Such*
*a twist-angle-dependent anisotropic SHG response demonstrates that the mirror symmetry at*
*MoS₂/SiP₂ heterointerface plays key role in generating anisotropic optical responses at the moiré*
*superlattice.*

As shown in the polarization-resolved PL results for 1L-MoS₂/SiP₂ heterostructures with various
 twist angles (Supplementary Figure 12a-f), two important features can be observed as follows: 1)
 the anisotropic PL responses exist at all twist angles; 2) the polarization degree ($P = \frac{I_{max}-I_{min}}{I_{max}+I_{min}}$,
 where I_{max} and I_{min} are the maximum and minimum PL intensity) of the PL, which can qualitatively
 reflect the anisotropic index of PL, shows a strong dependence with the twist angles. For example,
 the polarization degree P is 0.33 for 0° case and decreases to 0.14 with the twist angle increasing
 up to 30°. Such a decrease in the anisotropic index of PL might be related to the vanishing of the
 mirror symmetry of MoS₂/SiP₂ moiré superlattice plays an important role in controlling the magnitude of
 anisotropic optical responses at the heterointerface.”

**Figure R13 | Twist-angle-dependent anisotropic PL response of the 1L-MoS₂/SiP₂**
 **heterostructure. a-f,** Polarization-resolved PL integrated intensities of excitons in 1L-MoS₂/SiP₂
 with twisted angle of 0° (a), 19° (b), 27° (c), 33° (d), 37° (e), 50° (f). The solid circle is the
 experimental data, and the solid line is a fitting result using a $\cos^2\theta$ function. **g,** The polarization
 degree of the excitons in 1L-MoS₂/SiP₂ as a function of twist angle.

We also added the corresponding discussion sentences on page 9 of the revised manuscript:

*“More interestingly, the anisotropic PL and SHG responses strongly depended on the twist angle*
*between MoS₂ and SiP₂, and the corresponding anisotropy dramatically decreased when the mirror*
*symmetry of moiré superlattice vanishes with changing the twist angle (Supplementary Figs. 11 and*
*12). This result indicates that the mirror symmetry of MoS₂/SiP₂ moiré superlattice plays an*
*important role in controlling the magnitude of anisotropic optical responses at the heterointerface.”*

*1. The lack of surface morphology characterizations throughout the current manuscript is a concern.*
*The authors should provide morphological characterizations (e.g., via KPFM, PFM, or STM),*
*demonstrating the formation of anisotropic moiré superlattices at the interface of MoS₂ and SiP₂*
*with identified twist angles.*

**Authors’ response:**

We appreciate the reviewer for raising his/her concern about the morphological characterizations of
the anisotropic moiré superlattices at the 1L-MoS₂/SiP₂ heterointerface. We think that the atomic
morphological characterization to show anisotropic moiré superlattices at the interface of MoS₂ and
SiP₂ is technically super hard and might not be necessary at this stage for the following two reasons.

1) It has been proven that theoretical calculation is a powerful approach to simulate the moiré pattern
at the heterostructure for supporting the moiré-related phenomena in experiments. For example, in
the reports on unconventional superconductivity in twisted bilayer graphene²⁴, and the moiré exciton
in twisted TMDCs^{25,26} as well as the bulk photovoltaic effect in the BP/WSe₂ heterostructure⁶, no
direct experimental surface morphology characterization is applied. Instead, they use theoretical
calculations to simulate the moiré pattern therein and explain the related interfacial phenomena.
Similar to these works, we use DFT calculations to simulate the moiré pattern at the interface and
succeed in understanding the observed optical and electrical transport at the heterointerface in the
picture of the moiré physics.

2) Actually, the direct characterization of the surface morphology of the moiré superlattice via KPFM
or STM is a challenging task. It requires the ultra-high resolution (~Å) to observe the atomic stacking
in the moiré pattern. Such discovery should be an independent work and worthy to report in top
journals. For example, the reports on the atomic-resolution surface morphology characterization of
twisted bilayer graphene and twisted TMDCs were published in *Nature*²⁷⁻³¹. Therefore, it may not be
fair to require us to provide such a tough and top-level surface morphology characterization of the
moiré superlattice at this stage.

Although we think it may not be necessary to provide such a morphology characterization at this
stage since the current data are already convincing enough to explain the observed phenomena, we
still totally agree with the reviewer that the morphological characterizations of anisotropic moiré
superlattices can help understand the moiré physics therein and will be a good research topic in the
near future. By following the reviewer’s suggestion, we have performed KPFM measurements on
our MoS₂/SiP₂ heterostructure. However, atomically-resolved observation of the moiré superlattice
is still challenging due to the resolution limit of our KPFM equipment (Fig. R14).

To address the reviewer’s comment about the morphological characterizations of the interface, we
added the KPFM results of the MoS₂/SiP₂ heterostructure and corresponding discussions on page 38
of the revised Supplementary Information:

*“Importantly, the morphological characterizations of the heterostructures via KPFM have been*
*proven to be a powerful tool to directly observe the moiré superlattice therein. From this perspective,*
*the morphological characterizations of anisotropic moiré superlattices at the 1L-MoS₂/SiP₂ interface*
*with identified twist angles via KPFM can help understand the moiré physics therein and can be a*
*good research topic in the near future.”*

 **Figure R14 | KPFM measurements of the 1L-MoS₂/SiP₂ heterostructure.** **a**, Atomic force
 microscope image of the MoS₂/SiP₂ heterointerface. The regions of the monolayer MoS₂, Au film,
 1L-MoS₂/SiP₂ heterostructure and SiP₂ flake are labeled. **b**, KPFM imaging of the MoS₂/SiP₂
 heterointerface. The regions of the monolayer MoS₂, Au film, 1L-MoS₂/SiP₂ heterostructure and SiP₂
 flake are labeled.

*2. The symmetry of MoS₂/SiP₂ moiré superlattice, thus, the electronic band structure, optical*
 *responses, and electrical properties of the heterostructure depend on the twist angle between MoS₂*
 *and SiP₂. In this regard, the authors should prepare a few MoS₂/SiP₂ devices with varying twist*
 *angles, and interrogate the outcome, e.g., the polarization-resolved PL and SHG, and the*
 *conductance index.*

**Authors' response:**

We appreciate the reviewer for raising his/her concern about the twist angle dependent optical and
 electronic properties. The anisotropic PL and SHG responses indeed strongly depend on the twist
 angle between MoS₂ and SiP₂, and the corresponding anisotropy dramatically decreases when the
 mirror symmetry of the moiré superlattice vanishes with changing twist angles.

Specifically, in those 1L-MoS₂/SiP₂ heterostructures with different twist angles (for example 0° and
 50° shown in Fig. R12), the polarized SHG signals display the strong anisotropic SHG behavior,
 which is remarkably different from the six-fold symmetric SHG signals in pristine 1L-MoS₂. Such
 contrasting behavior before and after stacking SiP₂ to 1L-MoS₂ indicates that the C₃ symmetry of
 1L-MoS₂ is broken by the C₂ symmetry of the bottom SiP₂. More importantly, if we compare the
 polarized SHG signals between those cases with mirror symmetry and without mirror symmetry, one
 can see that the 0° case with mirror symmetry shows stronger anisotropy in SHG signals than that of
 the 50° case without mirror symmetry. Such a twist-angle-dependent anisotropic SHG response
 demonstrates that the mirror symmetry at MoS₂/SiP₂ heterointerface may play a key role in
 generating anisotropic optical responses at the moiré superlattice.

As shown in the polarization-resolved PL results for 1L-MoS₂/SiP₂ heterostructures with various
 twist angles (Fig. 13a-f), two important features can be observed as follows: 1) the anisotropic PL
 responses exist at all twist angles; 2) the polarization degree ($P = \frac{I_{\max} - I_{\min}}{I_{\max} + I_{\min}}$, where I_{\max} and I_{\min} are
 the maximum and minimum PL intensities) of the PL, which can qualitatively reflect the anisotropic
 index of PL, shows a strong dependence on the twist angle. For example, the polarization degree P
 is 0.33 for the 0° case and decreases to 0.14 as the twist angle increases to 30°. Such a decrease in
 the anisotropic index of PL might be related to the vanishing of the mirror symmetry with twist angle
 changing from 0° to 30°. This result indicates that the mirror symmetry of the MoS₂/SiP₂ moiré
 superlattice plays an important role in controlling the magnitude of anisotropic optical responses at
 the heterointerface.

Since the SHG and PL results have consistently demonstrated that the twist-angle-dependent
anisotropic optical properties at the heterostructure, similar twist-angle-dependent anisotropic
electronic properties (such as the anisotropic conductance index) at the interface can also be expected,
which can be a good research topic in the near future.

To address the reviewer's comment about twist-angle-dependent optical and electronic properties,
we added Figs. S11 and S12 and corresponding discussion on page 22 of the revised Supplementary
Information:

*“To investigate the effect of twist angle on the anisotropic properties of the heterostructure, we*
*performed polarization-resolved PL and SHG measurements on various 1L-MoS₂/SiP₂*
*heterostructures with different twist angles. Specifically, in those 1L-MoS₂/SiP₂ heterostructures with*
*different twist angles (for example 0° and 50° shown in Supplementary Fig. 11), the polarized SHG*
*signals displays the strong anisotropic SHG behavior, which is remarkably different from the six-fold*
*symmetric SHG signals in pristine 1L-MoS₂. Such contrasting behavior before and after stacking*
*SiP₂ to 1L-MoS₂ indicates that the C₃ symmetry of 1L-MoS₂ is broken by the C₂ symmetry of the*
*bottom SiP₂. More importantly, if we compare the polarized SHG signals between those cases with*
*mirror symmetry and without mirror symmetry, one can see that the 0° case with mirror symmetry*
*shows stronger anisotropy in SHG signals than that of the 50° case without mirror symmetry. Such*
*a twist-angle-dependent anisotropic SHG response demonstrates that the mirror symmetry at*
*MoS₂/SiP₂ heterointerface plays key role in generating anisotropic optical responses at the moiré*
*superlattice.*

*As shown in the polarization-resolved PL results for 1L-MoS₂/SiP₂ heterostructures with various*
*twist angles (Supplementary Figure 12a-f), two important features can be observed as follows: 1)*
*the anisotropic PL responses exist at all twist angles; 2) the polarization degree ($P = \frac{I_{max}-I_{min}}{I_{max}+I_{min}}$,*
*where I_{max} and I_{min} are the maximum and minimum PL intensity) of the PL, which can qualitatively*
*reflect the anisotropic index of PL, shows a strong dependence with the twist angles. For example,*
*the polarization degree P is 0.33 for 0° case and decreases to 0.14 with the twist angle increasing*
*up to 30°. Such a decrease in the anisotropic index of PL might be related to the vanishing of the*
*mirror symmetry with twist angle changing from 0° to 30°. This result indicates that the mirror*
*symmetry of MoS₂/SiP₂ moiré superlattice plays an important role in controlling the magnitude of*
*anisotropic optical responses at the heterointerface.”*

*3. With the particular crystal alignment between MoS₂ and SiP₂, the simulated results, e.g., in*
*Supplementary Figure 20, show visually isotropic moiré-modulated structural corrugations and*
*charge-density distributions, likely having a C₃ rotational symmetry. It is unclear how these visually-*
*isotropic simulated results corroborate the observed anisotropic conductance.*

**Authors' response:**

There might be a big misunderstanding here. Actually, the simulated results show clear anisotropic
(rather than isotropic) moiré-modulated structural corrugations and charge-density distributions with
mirror symmetry C_{1v} (rather than C_3 rotational symmetry), which can be clearly observed in a large
scale plot (see Fig. R15d). Such symmetry reducing will effectively influence the hopping between
the direction along and perpendicular to the mirror plane and eventually corroborate the observed
anisotropic conductance.

Please allow us to explain it in detail.

To clearly demonstrate the anisotropic moiré-modulated structural corrugations and charge-density
distributions at the interface, we show the real space distribution of the interlayer distance between
the SiP₂ layer and MoS₂ layer for the moiré pattern in Fig. R15b-d (Fig. 4c and Fig. S26 in the revised
manuscript). One can clearly see in Fig. R15d that, as shown in a wide range of moiré lattices, the
distribution of the interlayer distance (or the distribution of charge density) shows strong anisotropy
with the mirror symmetry parallel to the x direction of the heterostructure rather than the visual C_3 -
rotational symmetry.

To further demonstrate how the anisotropic moiré-modulated structure corroborates the observed
 anisotropic conductance, we analyze the corresponding charge-density distributions within the moiré
 superlattice. As shown in Fig. R15b and c, the stacked region of I-BA hosts the smallest interlayer
 distance and the largest charge density at the conduction band edge of the moiré superlattice, which
 indicates that the large interlayer potential in the I-BA region can effectively trap the charge carriers
 inside. Thus, the conductance of 1L-MoS₂ is mainly contributed by the effective hopping between
 those trapped charge states in the moiré potentials^{2,32-34}. Due to the anisotropic moiré potential at the
 interface, the effective hopping between the trapped charge states along different directions can be
 different and lead to the anisotropic conductance therein. For example, one can estimate reasonably
 that the effective hopping along the parallel direction (\parallel) to the mirror plane is naturally smaller than
 that along the perpendicular direction (\perp), indicating that, at the off state, the effective mass of these
 trapped states is highly anisotropic ($m_{\parallel} \gg m_{\perp}$). Such a large ratio of effective mass (m_{\perp}/m_{\parallel}) leads
 to highly anisotropic conductance in the 1L-MoS₂/SiP₂ heterostructure.

**Figure R15 | The simulations of the structural corrugations and the charge densities at the**
 **conduction band edge of the commensurate moiré superlattices of case-I.** **a**, The top view of the
 moiré superlattice of case-I. **b**, The simulation of the structural corrugations (interlayer distance
 between SiP₂ and MoS₂) of the moiré superlattice of case-I. **c**, The charge density at the conduction
 band edge of the moiré superlattice of case-I. **d**, The real space distribution of the interlayer distance
 in the moiré pattern of case-I. The dashed rectangular area corresponds to the moiré superlattice
 shown in (a) and (b).

To address the reviewer's comment about the visually isotropic moiré-modulated structural
 corrugations, we revised the sentences on page 49 of the revised Supplementary Information: "*One*
 *can clearly see that the distribution of the interlayer distance (or the distribution of charge density)*
 *shows strong anisotropy with the mirror symmetry parallel to the x direction of the heterostructure*
 *(more clearly in Fig. 4 with a wide range of moiré lattices).* Thus, the anisotropic moiré potential in
 the 1L-MoS₂/1L-SiP₂ heterointerface can be described by the structural fluctuation and the real

*space charge density distribution of the conduction band edge. The states trapped by moiré potential*
*mainly stay at the atomic interface in MoS₂/SiP₂ heterostructure. Such anisotropic moiré potential*
*influence the hopping between the direction along and perpendicular to the mirror plane and*
*eventually corroborate the observed anisotropic conductance.”*

*4. To identify the hopping path, the moiré potential should be mapped.*

**Authors' response:**

We appreciate the reviewer for raising his/her concern about the moiré potential at the MoS₂/SiP₂
heterointerface. Following the methods used in Ref. ³⁵, the moiré potentials of case-I and case-II are
shown in Fig. R16. The patterns of the calculated moiré potential are exactly the same as those for
the structural corrugations in 1L-MoS₂/SiP₂ heterostructure.

In traditional moiré systems, such as twisted bilayer graphene (TBG)³⁶ and twisted bilayer transition
metal dichalcogenides (TB-TMDC)^{37,38}, there is a small lattice mismatch between the adjacent layers.
With the small twist angle, the moiré potential can be calculated based on the potential at each
stacking configuration. Thus, once we know the electronic structures in these stacking configurations,
such as AA, AB and BA, we could obtain the moiré potential via building several unit-cell models
that contain different stackings and performing the DFT calculations. Such a method has been widely
used to calculate the moiré potential in TBG and TB-TMDC.

However, we cannot apply this widely-used method to 1L-MoS₂/SiP₂ heterostructure directly. For
1L-MoS₂/SiP₂ heterostructure, the lattice constants a for 1L-MoS₂ and 1L-SiP₂ are 3.16 Å and 3.44
Å, and the lattice mismatch is about 9%. Such a large lattice mismatch can induce the moiré patterns
even without twisting. If we still apply the method mentioned above to calculate with the unit-cell,
the artificial strain will completely change the electronic structures of 1L-MoS₂ and 1L-SiP₂.
Therefore, the aforementioned method is not suitable for the current case and a new method is
required.

In order to simulate the moiré potential in 1L-MoS₂/SiP₂ heterostructure correctly, we use the method
introduced in Ref. ³⁵. They considered a moiré system constituted by the 1L-MoTe₂ (9 × 9) and 1L-
MoS₂ (10 × 10). The authors directly extracted the valence band edge (VBM) and the conduction
band edge (CBM) at the AA, AB, and BA stackings from the large moiré pattern calculations.
Explicitly, the values of VBM and CBM of AA, AB, and BA stackings are estimated by the analysis
of the logarithm of the local density of states (LDOS) projected on Mo atoms at each high-symmetry
stacking area. Using the same method, the mappings of the moiré potential of case-I and case-II
heterostructures are shown in Fig. R16c and f.

On the other hand, we need to point out that the calculated energy potential extracted from LDOS is
very sensitive to the parameter setting, such as the smearing, used in DFT calculations. We cannot
fully trust the calculated moiré potential value. While, based on our experience, the mappings of the
interlayer distance are more reliable, because the influence of the smearing is relatively trivial during
the structural relaxation. This is also the reason that we chose to illustrate the hopping model with
the mappings of interlayer distance instead of the mappings of moiré potential in the revised
manuscript.

**Figure R16 | The simulations of the structural corrugations and the moiré potentials of the**
 **commensurate moiré superlattices of case-I and case-II. a,** The top view of the moiré superlattice
 of case-I. **b,** The simulation of the structural corrugations of the moiré superlattice of case-I. **c,**
 the mapping of the moiré potential in the moiré superlattice of case-I. **d,** The top view of the moiré
 superlattice of case-II. **e,** The simulation of the structural corrugations of the moiré superlattice of
 case-II. **f,** The mapping of the moiré potential in the moiré superlattice of case-II.

*Other comments to clarify:*

*1. On page 4-5, the authors contrasted the gate voltages for SiO₂ (80V) with that for SiP₂ (5V) when*
 *achieving a similar on/off current ratio. The current should be related to the doping density, and the*
 *doping density $n \cdot e$ is generally given by $n \cdot e \approx C_{top} \cdot V_{ig} + C_{bot} \cdot V_{bg} \approx (\epsilon_{top} \cdot A / d_{top}) V_{ig} + (\epsilon_{bot} \cdot A / d_{bot}) V_{bg}$,*
 *where e is the electron charge, C_{top} (C_{bot}) is the capacitance associated with the top (bottom)*
 *gate, the top (bottom) dielectric medium, and MoS₂; ϵ_{top} (ϵ_{bot}) is the dielectric constant of the top (bottom)*
 *gating dielectric medium; d_{top} (d_{bot}) is the distance between the top (bottom) gate and MoS₂; and V_{ig}*
 *(V_{bg}) is top (bottom) gate voltage. In this regard, the contrasting gate voltages required to achieve*
 *the same on/off are naturally determined by the different thicknesses and dielectric constants of the*
 *gating medium. It is unclear what the authors were trying to emphasize on page 4-5 when describing*
 *the contrasting gate voltages.*

**Authors' response:**

We appreciate the reviewer for raising his/her concern. When describing the contrasting gate voltages
 on pages 4-5 of the manuscript, we try to emphasize the great capacitive capability of the SiP₂ gate

dielectric and its large dielectric breakdown field (even with small thickness) for achieving low-
voltage operations in devices. As noticed by the reviewer, the smaller gate voltages in SiP₂-gated
MoS₂ FETs is a direct result of the larger dielectric constant and smaller thickness (without dielectric
breakdown) in the SiP₂ gate medium. In fact, a larger dielectric constant and a smaller thickness
(without dielectric breakdown) in gate mediums are highly demanded for low-voltage operations.
For example, one needs to use 10-nm SiO₂ as the gate medium to realize an on/off ratio similar to
that of our 20-nm SiP₂; however, 10-nm SiO₂ is hard to fabricate and easily leaks in practical
applications (300 nm SiO₂ is widely used as the gate medium in practical electronic applications).
Therefore, a direct comparison of the 20-nm-SiP₂-gated FETs with the widely used 300-nm-SiO₂-
gated FETs can directly show the great dielectric properties of SiP₂ for achieving low-voltage
operations in devices.

To clarify such an issue and to make a better understanding, we revised the corresponding sentences
on page 4 of the revised manuscript: “*In contrast, when sweeping the bottom gate voltage V_{bg-SiO_2}
to ~ 5 V, the transistor generates an on/off ratio as small as 10 and requires a huge V_{bg-SiO_2} over 80
V to achieve an on/off ratio of 10^5 (inset of Fig. 1c). This comparison directly demonstrates that SiP₂
gate medium with larger dielectric constant and smaller thickness can achieve great capacitive
capability.*”

*2. On page 5, the authors mentioned the mobility of MoS₂ in a SiP₂-gated device vs. that of MoS₂ in
an HfO₂-gated device. Since mobility μ is derived from $\mu = e\tau/(m)$, where e is an electron charge, τ
is the carrier scattering time, and m is the carrier's effective mass, why the mobility of MoS₂ depends
on the gating medium? Alternatively, does this difference come from device variations?*

**Authors' response:**

We appreciate the reviewer for raising his/her concern on the factors influencing the mobility. Since
the carrier scattering time τ at the semiconductor/dielectric interface can be remarkably different by
changing gate medium, the mobility of ultrathin MoS₂ can eventually depend on the gating medium.

In general, by changing the gate medium, there are two mechanisms to modify the charge scattering
and thus the mobility of the semiconductor: 1) Using high- κ gate mediums to change the dielectric
environment and effectively screen the Coulomb scattering in semiconductors: for example, high- κ
HfO₂ is reported as the top-gate dielectric to reduce charge scattering and results in mobility
improvement of MoS₂ FETs (if compared to SiO₂-gated MoS₂ FETs)². 2) The utilization of the vdW
dielectric as a gate medium to form an atomically flat interface with the semiconductor layer can
reduce charge scattering and enhance device mobility: for example, vdW dielectric *h*-BN is reported
to reduce charge scattering and result in a giant mobility improvement in MoS₂ FETs ($1000 \text{ cm}^2 \text{ V}^{-1}$
s^{-1} at low temperature)^{39,40}. As indicated in the comparison of 1L-MoS₂ FETs gated with different
dielectrics (Table R1), the mobility level of 1L-MoS₂ FETs always depends on the dielectrics.

As a typical vdW material with a relatively large dielectric constant, SiP₂ can not only effectively
screen the Coulomb scattering (as those high- κ gate mediums), but also form atomically flat
interfaces with the semiconductor layer (as those vdW gate mediums) to reduce charge scattering
and increase the mobility of MoS₂ FETs. Therefore, the difference in the mobility in the HfO₂-gated
device and SiP₂-gated device does not come from device variations and is simply because these two
gate media have different abilities to reduce charge scattering at the dielectric/semiconductor
interface.

To clarify the mobility issue on SiP₂-gated MoS₂ FETs, we revised the corresponding sentences page
5 of the revised manuscript: “*Even for SiP₂-gated 1L-MoS₂ transistors with the same device geometry
(Supplementary Fig. 2), the mobility of $330 \text{ cm}^2 \text{ V}^{-1} \text{ s}^{-1}$ at 2 K is better than those reported values in
HfO₂-gated 1L-MoS₂ devices² ($174 \text{ cm}^2 \text{ V}^{-1} \text{ s}^{-1}$ at 4 K), indicating that the vdW SiP₂ material with a
large dielectric constant can effectively reduce the charge scattering and increase the mobility of
MoS₂ transistors.*”

Table R1. Comparison of 1L-MoS₂ FETs gated with different dielectrics

Dielectrics	On/off ratio	Mobility (cm ² V ⁻¹ s ⁻¹)	E_{bd} (MV cm ⁻¹)	Operation voltage (V)	Refs.
SiP ₂	10 ⁵	300 (2 K)	> 2	< 5	This work
SiO ₂	10 ³ –10 ⁵	~1 (300 K) ~20 (40 K)	> 3	~ 80	41,42
HfO ₂	10 ⁸	174 (4 K)	> 2	< 5	2
Al ₂ O ₃	10 ⁷	28 (300 K)	> 4.5	< 3	43
h -BN	10 ⁵	60 (300 K) 1000 (5 K)	> 2	< 4	39,44
Sb ₂ O ₃	10 ⁸	80 (40 K)	1.8	> 20	42
FS-SrTiO ₃	10 ⁷	40 (300 K)	> 2	< 2	45
CaF ₂	10 ⁴ –10 ⁷	NA	10–15	< 2	46

*3. What are the x and y directions in Figure 2a; how were the crystal axes of MoSe₂ and SiP₂*
 *experimentally confirmed in the optical image?*

**Authors' response:**

We appreciate the reviewer's concern about the labeling the crystal direction. We have labeled the *x*
 and *y* (zigzag and armchair direction of the MoS₂) directions in Fig. 2a of the revised manuscript. In
 our experiments, the crystal axes of MoS₂ are confirmed by the polarized SHG results, and the crystal
 axes of SiP₂ are confirmed by the polarized PL results. We characterized all the sample directions
 before performing the optical and transport measurements.

For example, in the case of the 1L-MoS₂ sample, we performed polarized-SHG measurements under
 a parallel measurement geometry to confirm its crystal axes. As shown in Fig. R17a, the SHG signal
 in 1L-MoS₂ shows a sixfold-rotational symmetric pattern, and the maxima of SHG intensity appear
 along its armchair direction^{47,48} (the zigzag direction perpendicular to the armchair direction can be
 determined correspondingly). While for the SiP₂ sample, since the SiP₂ flake has inversion
 symmetry⁴⁹ and shows no obvious SHG signals, we performed polarized PL measurements to
 confirm its crystal axes, which have been widely used in previous reports^{49,50}. As shown in Fig. R17b,
 the PL signals in the SiP₂ flake show a twofold-rotational symmetric pattern, and the maxima of the
 PL intensity appear along its *x* direction (the *y* direction perpendicular to the *x* direction can be
 determined correspondingly). After determining the crystal axes of the two materials, we used the
 dry transfer method to fabricate heterostructures with defined twist angles.

To address the *x/y* labeling issue on the crystal axes, we replotted Fig. 2a to add the *x* and *y* direction
 labels and added several sentences on page 16 in the method section in the revised manuscript: "*The*
 *crystal axes of the 1L- MoS₂ samples are confirmed by their polarized-SHG results. And the crystal*
 *axes of SiP₂ are first identified by their optical image and then confirmed by their polarized-PL*
 *results.*"

 **Figure R17. Determination of the crystal axes of 1L-MoS₂ and SiP₂ flakes.** **a**, Polar plot of
 polarization-resolved SHG intensities of 1L-MoS₂ under the parallel configuration (the detection
 polarization is parallel to the excitation polarization). The red solid line represents the fitting and the
 blue balls represent the experimental data. **b**, Polar plots of polarization-resolved PL integrated
 intensities of SiP₂ (red curve). The purple balls represent the experimental data and the red solid
 curves represent the sine fitting.

*4. How do the SHG anisotropy in Figure 2c and the PL polarization in Figure 2f orient on top of the*
 *optical image in Figure 2a? Were the SHG and PL taken from one spot, or multiple spots in the*
 *heterostructures? How about the consistency?*

**Authors' response:**

We appreciate the reviewer for raising his/her concern on the consistency of the SHG and PL results
 of the heterostructure. For polarized-SHG and PL measurements, the data were taken from one spot
 for a strict comparison. Importantly, we performed SHG and PL measurements on various
 TMDC/SiP₂ heterostructures to confirm the excellent reproducibility and consistency of the results
 in different samples. We also performed PL mapping measurements at multiple spots throughout a
 region to confirm the consistency of the result at multiple spots of a single TMDC/SiP₂
 heterostructure. All the results are consistent with each other.

To confirm the consistency of the SHG and PL results on various TMDC/SiP₂ heterostructures, we
 performed SHG and PL measurements on more than 8 TMDC/SiP₂ heterostructures. As shown in
 Fig. R18, all samples show similar anisotropic SHG and PL results, indicating the great
 reproducibility and consistency of the results. In particular, the SHG results obtained from different
 spots at the same 1L-WSe₂/SiP₂ heterostructure are almost the same (Fig. R18e,f), demonstrating the
 great consistency of the anisotropic SHG response.

To further confirm the consistency of the result at multiple spots of a single TMDC/SiP₂
 heterostructure, we performed PL mapping on a 1L-MoS₂/SiP₂ heterostructure under different
 polarization configurations (detection polarizations along the *x* and *y* directions). As shown in Fig.
 R19, the exciton emission exhibits a sharp contrast in the PL intensities between the *x* and *y* detection
 polarizations. More importantly, the PL intensity crossing the whole mapping region at a fixed
 detection angle is modestly uniform, which suggests that partially linearly-polarized exciton states
 and symmetry engineering uniformly exist in the heterostructure. Such a result directly shows the
 great consistency of the anisotropic optical results at multiple spots of the heterostructure.

**Figure R18 | Reproducibility of anisotropic SHG and PL in SiP₂/1L-TMDC heterostructures.**
 **a-f**, Polar plot of polarization-resolved SHG intensities of SiP₂/1L-TMDC heterostructures under the
 parallel configuration (the detection polarization is parallel to the excitation polarization). The red
 solid line represents the fitting and the blue balls represent the experimental data. The insets are the
 corresponding optical images for the heterostructures. The red cross highlights the measured position.
 **g-i**, Polar plots of polarization-resolved PL integrated intensities of SiP₂/1L-TMDC heterostructures.
 The purple balls represent the experimental data and the red solid curves represent the sine fitting.
 The insets are the corresponding optical images for the heterostructures. The red cross highlights the
 measured position.

**Figure R19 | Polarization-dependent PL in 1L-MoS₂/SiP₂.** **a**, Schematic of the polarized PL
 measurement geometry. The polarization of the incident laser is fixed along the *x* direction, while the
 emission light is collected by the analyzer with various polarizations. **b**, PL spectra of 1L-MoS₂/SiP₂
 with an analyzer along the *x* and *y* directions. **c**, Schematic (top) and optical image (bottom) of the
 1L-MoS₂/SiP₂ heterostructure with the alignment of the armchair direction of MoS₂ and the *x*
 direction of SiP₂. The red (or white) dashed line outlines the edge of the MoS₂ (or SiP₂) sample. The
 dashed lines represent the mirror plane. Scale bar is 20 μm. **d**, Polarization-resolved PL integrated
 intensities of the exciton peak in 1L-MoS₂/SiP₂. The solid line is the fitting result using a cos²θ
 function, in which θ denotes the angle between the detection polarization and the *x* direction. **e**,
 Schematic of the polarized PL mapping measurement geometry with the analyzer along the *x* and *y*
 directions. **f**, PL mapping of the intensities of the exciton peak in MoS₂ at 77 K with an analyzer
 along the *x* and *y* directions. The PL mapping area corresponds to the green square in (c). Scale bar
 is 1 μm.

To address the reviewer’s comment on the consistency of the SHG and PL results of the
 heterostructure, we added new optical results on various samples and corresponding discussion on
 page 16 of the revised Supplementary Information:

*“To further confirm the consistency of the anisotropic PL response at multiple spots of the*
 *TMDC/SiP₂ heterostructure, we performed PL mapping on a 1L-MoS₂/SiP₂ heterostructure with*
 *detection polarizations along the *x* and *y* directions (Supplementary Fig. 6d–f). Both peak 1 and peak*
 *2 exhibit a sharp contrast in the PL intensities between the *x* and *y* detection polarizations.*
 *Furthermore, the PL intensity crossing the whole mapping region at a fixed detection angle is*
 *modestly uniform, which suggests that partially linearly polarized exciton states and symmetry*
 *engineering uniformly exist in the heterostructure. Such a result directly shows the great consistency*
 *of the anisotropic PL responses at multiple spots of the heterostructure.”*

Regarding the reviewer’s comment on *SHG anisotropy in Figure 2c and the PL polarization in*
 *Figure 2f orient on top of the optical image in Figure 2a*, both the strongest SHG and PL intensities
 appear at θ = 0°, in which θ denotes the angle between the analyzer polarization direction and the *x*
 direction in Fig. 2a (θ = 0° represents the *x* direction).

To address the reviewer's comment on the polarization of the SHG and PL response, we replotted
 Fig. 2a to add the x and y direction labels in the revised manuscript.

*5. The microscopic origin of the polarized PL is unclear. The nature of the exciton should be related*
 *to the Wannier states confined by the moiré potential (likely at the AA or BA site based on*
 *Supplementary Figure 20c and f); how is the optical selection rule derived from this, and how this is*
 *related to the PL polarization?*

**Authors' response:**

We appreciate the reviewer for raising his/her concern about the microscopic origin of the polarized
 PL. In fact, herein, we could understand this issue from two physical pictures, which are consistent
 and equivalent with each other. The first one is mentioned by the reviewer, we could start the whole
 story from the Wannier excitons in TMDC first and regard the influence of the moiré potential from
 SiP₂/TMDC interface as an external potential to modify the property of the Wannier exciton. In such
 cases, the Hamiltonian of the moiré exciton has the form in the basis of the TMDC Wannier excitons

$$1093 \quad H^{\text{moiré}} = \hbar\Omega_0 + \frac{\hbar^2 Q^2}{2M} + V_{\text{moiré}}$$

Herein, \hbar is the reduced Planck constant, $\hbar\Omega_0$ is the energy of TMDC Wannier excitons, Q and M
 are the momentum and effective mass of the Wannier excitons, and $V_{\text{moiré}}$ is the moiré potential. The
 second term stands for the kinetic energy of the Wannier excitons. Thus the eigenstate of moiré
 excitons $|\psi_{ex}^{\text{moiré}}\rangle$ satisfies

$$1098 \quad H^{\text{moiré}}|\psi_{ex}^{\text{moiré}}\rangle = E^{\text{moiré}}|\psi_{ex}^{\text{moiré}}\rangle$$

with the form of $|\psi_{ex}^{\text{moiré}}\rangle = A^{\text{moiré}}(r)a_e^\dagger a_h|0\rangle$, in which $E^{\text{moiré}}$ is the energy of moiré excitons,
 $A^{\text{moiré}}(r)$ is the wavefunction of moiré excitons modified by the moiré potential, a_e^\dagger and a_h are the
 creation and annihilation operators for states at the conduction and valence band edges of TMDC
 without moiré potential, and $|0\rangle$ is for the charge neutral ground state. Influenced by the moiré
 potential, the $A^{\text{moiré}}(r)$ should have the same symmetry as the moiré potential (C_{1v} in our case). Due
 to the Fermi golden rule, the absorption of moiré excitons $I^{\text{moiré}}$ is determined by the optical matrix
 elements $I^{\text{moiré}} \sim |\langle 0|H_{LM}|\psi_{ex}^{\text{moiré}}\rangle|^2$, thus we have

$$1106 \quad I^{\text{moiré}} \sim |\langle 0|H_{LM}A^{\text{moiré}}(r)a_e^\dagger a_h|0\rangle|^2 = \left(A^{\text{moiré}}(r)\right)^2 |\langle 0|H_{LM}a_e^\dagger a_h|0\rangle|^2$$

$$1107 \quad = \left(A^{\text{moiré}}(r)\right)^2 |\langle h|H_{LM}|e\rangle|^2$$

Herein, H_{LM} stands for the light-matter interactions, $|e\rangle$ and $|h\rangle$ are for the electron state and hole
 state of TMDC without the moiré potential. Thus, the absorption of moiré excitons is determined by
 the square of the moiré exciton wavefunction and the absorption of intrinsic TMDC. Because the
 symmetry of the moiré potential is lower than that of intrinsic TMDC, the symmetry of $I^{\text{moiré}}$ is
 same with that of moiré potential. Furthermore, the PL spectrum is determined by $I^{\text{moiré}}$ in the lowest
 bright excitonic state. Thus, the PL spectrum has the same symmetry with the moiré potential.

On the other hand, in the second physical picture, we could consider the influence of the moiré
 potential on the electronic structures for the conduction and valence band edges firstly, then take into
 account the excitonic excitation. Under the moiré potential, the wavefunction for the conduction and
 valence band edge of TMDC $|\psi_{c(v)-m}\rangle$ will be strongly modulated, its symmetry will be the same
 with that of moiré potential (see Fig. R20 for the charge density plot, corresponding to the sum of
 square of wavefunction $\sum_{x,z} \psi_{c-m}^2(\mathbf{r})$). In such case, the absorption $I^{\text{moiré}} \sim |\langle \psi_{c-m}|H_{LM}|\psi_{v-m}\rangle|^2$
 will be also modulated by the moiré potential.

**Figure R20 | Visualizations of the charge density on the conduction band edge in the real space**
 **and the corresponding plane-averaged charge density along the y direction of the strained slab**
 **models. a,** The visualizations of the charge density distribution in the real space (bottom panel) and
 the plane-averaged charge density along the y direction (top panel) of 1L-MoS₂. The iso-surface is
 set as 0.0001 e Å⁻³. **b,** The illustration of the interface of MoS₂/SiP₂ with the mirror symmetry. **c-f,**
 The visualizations of the charge density distribution in the real space (bottom panel) and the
 plane-averaged charge density along the y direction (top panel) of case-I-ABBA, case-II-AABA, case-I-
 AA, and case-II-AB, respectively. The iso-surface is set as 0.0002 e Å⁻³.

To address the reviewer's comment, we revised the corresponding discussion sentences on Page 13
 of the revised manuscript: "Therefore, when one electron is excited on conduction band edge and
 couples with hole states, the optical matrix elements in formed exciton should be strongly modified
 by the moiré potential with lower symmetry and the lowest bright exciton absorption becomes highly
 anisotropic, which is consistent with the observation from our PL experiments."

*6. If moiré potentials were formed at the interface of MoS₂/SiP₂, the doped charges were likely*
 *localized/trapped near the interface of MoS₂/SiP₂ in bulk MoS₂ due the moiré traps; how is the*
 *charge wavefunction spatial distribution in the bulk-MoS₂/SiP₂ different from that of the Monolayer-*
 *MoS₂/SiP₂, and how does this difference mechanistically give rise to different transport properties?*

**Authors' response:**

We appreciate the reviewer for raising his/her concern about the difference in the charge
 wavefunction spatial distribution in the bulk-MoS₂/SiP₂ and 1L-MoS₂/SiP₂ and the corresponding
 influence on the transport properties. First, we would like to point out that the anisotropic transport
 in the MoS₂/SiP₂ heterostructure can only be observed in low gate voltages and at low temperature.
 Such fact indicates that the trapping of electrons by moiré potential only plays an important role at
 low doping level (~5×10⁹ cm⁻² estimated from transport experiment) and only a few electrons are
 trapped in the moiré potential. In such case, we estimate that doped carriers only stay at the atomic
 interface. On the other hand, both MoS₂ and SiP₂ are layered materials, the van der Waals interaction
 binds the adjacent layers, the interlayer coupling is weak. Considering that the moiré potential is also
 an interface phenomenon, we expect the charge distribution of doped charges in bulk-MoS₂/SiP₂ to

be qualitatively similar to that in 1L-MoS₂/SiP₂.

Such effects can also be confirmed from the transport properties in the MoS₂/SiP₂ heterostructures
once we change the thickness of MoS₂. Specifically, the anisotropy index of MoS₂ can be written as
$\frac{G_y}{G_x} = \frac{G_y^{\text{surface}} + G_y^{\text{bulk}}}{G_x^{\text{surface}} + G_x^{\text{bulk}}}$, where bulk conductance (proportional to the sample thickness) is isotropic G_y^{bulk}
$\approx G_x^{\text{bulk}}$ while surface conductance is anisotropic since the doped charges are mainly trapped by the
moiré potential at the surface layer of MoS₂ with a certain thickness.

For the SiP₂-gated 1L-MoS₂ case, $G_y^{\text{bulk}} = G_x^{\text{bulk}} = 0$ and only the MoS₂ layer on the interface
(namely the whole monolayer) contributes to the conductance, so the anisotropy index can be written
as $\frac{G_y}{G_x} = \frac{G_y^{\text{surface}}}{G_x^{\text{surface}}}$, whose value can be as high as 1000. When increasing the thickness of MoS₂,
although the surface MoS₂ layer still shows strong anisotropic conductance, the bulk conductance
G_y^{bulk} and G_x^{bulk} begin to increase and gradually dominate the total conductance with $G^{\text{surface}} \ll$
G^{bulk} at the bulk limit. As a result, the anisotropy index is reduced to $\frac{G_y}{G_x} = \frac{G_y^{\text{bulk}}}{G_x^{\text{bulk}}} = 1$, which is
consistent with our experimental observation of less anisotropy in SiP₂-gated MoS₂ with a thickness
of 20 nm. To conclude, we have discussed the difference in the charge spatial distribution in bulk-
MoS₂/SiP₂ and 1L-MoS₂/SiP₂ and how these differences correspondingly influence their transport
properties.

To address the reviewer's comment on the charge spatial distribution in the bulk-MoS₂/SiP₂ and 1L-
MoS₂/SiP₂, we added several discussion sentences on page 49 of the revised Supplementary
Information: "*The states trapped by moiré potential mainly stay at the atomic interface in MoS₂/SiP₂*
*heterostructure. Such anisotropic moiré potential influences the hopping between the direction along*
*and perpendicular to the mirror plane and eventually corroborates the observed anisotropic*
*conductance.*"

In short, we have addressed all the reviewer's comments and suggestions point by point. We hope to
convince the reviewer that our revised manuscript meets the criteria for publication in *Nature*
*Communications*.

**References:**

- Sucharitakul, S. *et al.* V₂O₅: A 2D van der Waals Oxide with Strong In-Plane Electrical and
Optical Anisotropy. *ACS Appl. Mater. Interfaces* **9**, 23949–23956 (2017).
- Radisavljevic, B. & Kis, A. Mobility engineering and a metal-insulator transition in
monolayer MoS₂. *Nat. Mater.* **12**, 815–820 (2013).
- Li, Y. *et al.* Enhanced bulk photovoltaic effect in two-dimensional ferroelectric CuInP₂S₆.
*Nat. Commun.* **12**, 5896 (2021).
- Osterhoudt, G. B. *et al.* Colossal mid-infrared bulk photovoltaic effect in a type-I Weyl
semimetal. *Nat. Mater.* **18**, 471–475 (2019).
- Glass, A. M., von der Linde, D. & Negran, T. J. High-voltage bulk photovoltaic effect and
the photorefractive process in LiNbO₃. *Appl. Phys. Lett.* **25**, 233–235 (2003).
- Akamatsu, T. *et al.* A van der Waals interface that creates in-plane polarization and a
spontaneous photovoltaic effect. *Science* **372**, 68–72 (2021).
- Morimoto, T. & Nagaosa, N. Topological nature of nonlinear optical effects in solids. *Sci.*
*Adv.* **2**, e1501524 (2016).
- Sodemann, I. & Fu, L. Quantum Nonlinear Hall Effect Induced by Berry Curvature Dipole

in Time-Reversal Invariant Materials. *Phys. Rev. Lett.* **115**, 216806 (2015).
9 Ma, Q. *et al.* Observation of the nonlinear Hall effect under time-reversal-symmetric
conditions. *Nature* **565**, 337–342 (2019).
Sinha, S. *et al.* Berry curvature dipole senses topological transition in a moiré superlattice.
*Nat. Phys.* **18**, 765–770 (2022).
Xu, S.-Y. *et al.* Electrically switchable Berry curvature dipole in the monolayer topological
insulator WTe₂. *Nat. Phys.* **14**, 900–906 (2018).
Sodemann, I. & Fu, L. Quantum Nonlinear Hall Effect Induced by Berry Curvature Dipole
in Time-Reversal Invariant Materials. *Phys. Rev. Lett.* **115**, 216806 (2015).
Duan, S. *et al.* Berry curvature dipole generation and helicity-to-spin conversion at
symmetry-mismatched heterointerfaces. *Nat. Nanotechnol.*, doi:10.1038/s41565-023-01417-
z (2023).
Yu, Y. *et al.* Gate-tunable phase transitions in thin flakes of 1T-TaS₂. *Nat. Nanotechnol.* **10**,
270–276 (2015).
Liu, C. *et al.* Two-dimensional superconductivity and anisotropic transport at KTaO₃ (111)
interfaces. *Science* **371**, 716–721 (2021).
Illarionov, Y. Y. *et al.* Insulators for 2D nanoelectronics: the gap to bridge. *Nat. Commun.* **11**,
338 (2020).
Park, J. H. *et al.* Atomic Layer Deposition of Al₂O₃ on WSe₂ Functionalized by Titanyl
Phthalocyanine. *ACS Nano* **10**, 6888–689 (2016).
Li, T. *et al.* A native oxide high- κ gate dielectric for two-dimensional electronics. *Nat.*
*Electron.* **3**, 473–47 (2020).
Refaely-Abramson, S., Qiu, D. Y., Louie, S. G. & Neaton, J. B. Defect-Induced Modification
of Low-Lying Excitons and Valley Selectivity in Monolayer Transition Metal
Dichalcogenides. *Phys. Rev. Lett.* **121**, 167402 (2018).
Jeong, T. Y. *et al.* Spectroscopic studies of atomic defects and bandgap renormalization in
semiconducting monolayer transition metal dichalcogenides. *Nat. Commun.* **10**, 3825 (2019).
Greben, K., Arora, S., Harats, M. G. & Bolotin, K. I. Intrinsic and Extrinsic Defect-Related
Excitons in TMDCs. *Nano Lett.* **20**, 2544–255 (2020).
Vancso, P. *et al.* The intrinsic defect structure of exfoliated MoS₂ single layers revealed by
Scanning Tunneling Microscopy. *Sci. Rep.* **6**, 29726 (2016).
Tongay, S. *et al.* Defects activated photoluminescence in two-dimensional semiconductors:
interplay between bound, charged, and free excitons. *Sci. Rep.* **3**, 2657 (2013).
Cao, Y. *et al.* Unconventional superconductivity in magic-angle graphene superlattices.
*Nature* **556**, 43–50 (2018).
Seyler, K. L. *et al.* Signatures of moire-trapped valley excitons in MoSe₂/WSe₂ heterobilayers.
*Nature* **567**, 66–70 (2019).
Tran, K. *et al.* Evidence for moire excitons in van der Waals heterostructures. *Nature* **567**,
71–75 (2019).
Nuckolls, K. P. *et al.* Strongly correlated Chern insulators in magic-angle twisted bilayer
graphene. *Nature* **588**, 610–615 (2020).
Gadelha, A. C. *et al.* Localization of lattice dynamics in low-angle twisted bilayer graphene.
*Nature* **590**, 405–40 (2021).
Oh, M. *et al.* Evidence for unconventional superconductivity in twisted bilayer graphene.
*Nature* **600**, 240–245 (2021).
Kim, H. *et al.* Evidence for unconventional superconductivity in twisted trilayer graphene.
*Nature* **606**, 494–500 (2022).
Zhang, Z. *et al.* Flat bands in twisted bilayer transition metal dichalcogenides. *Nat. Phys.* **16**,
1093–1096 (2020).
32 Ma, N. & Jena, D. Charge Scattering and Mobility in Atomically Thin Semiconductors. *Phys.*
*Rev. X* **4**, 011043 (2014).
Kaasbjerg, K., Thygesen, K. S. & Jauho, A.-P. Acoustic phonon limited mobility in two-
dimensional semiconductors: Deformation potential and piezoelectric scattering in

monolayer MoS₂ from first principles. *Phys. Rev. B* **87**, 235312 (2013).
Kaasbjerg, K., Thygesen, K. S. & Jacobsen, K. W. Phonon-limited mobility in n-type single-
layer MoS₂ from first principles. *Phys. Rev. B* **85**, 115317 (2012).
Geng, W. T., Wang, V., Liu, Y. C., Ohno, T. & Nara, J. Moiré Potential, Lattice Corrugation,
and Band Gap Spatial Variation in a Twist-Free MoTe₂/MoS₂ Heterobilayer. *J. Phys. Chem.*
*Lett.* **11**, 2637–2646 (2020).
Yoo, H. *et al.* Atomic and electronic reconstruction at the van der Waals interface in twisted
bilayer graphene. *Nat. Mater.* **18**, 448–453 (2019).
Shabani, S. *et al.* Deep moiré potentials in twisted transition metal dichalcogenide bilayers.
*Nat. Phys.* **17**, 720–725 (2021).
Naik, M. H. & Jain, M. Ultraflatbands and Shear Solitons in Moiré Patterns of Twisted
Bilayer Transition Metal Dichalcogenides. *Phys. Rev. Lett.* **121**, 26640 (2018).
Cui, X. *et al.* Multi-terminal transport measurements of MoS₂ using a van der Waals
heterostructure device platform. *Nat. Nanotechnol.* **10**, 534–540 (2015).
Dean, C. R. *et al.* Boron nitride substrates for high-quality graphene electronics. *Nat.*
*Nanotechnol.* **5**, 722–726 (2010).
Han, T. *et al.* Probing the Field-Effect Transistor with Monolayer MoS₂ Prepared by APCVD.
*Nanomaterials* **9** (2019).
Liu, K. *et al.* A wafer-scale van der Waals dielectric made from an inorganic molecular crystal
film. *Nat. Electron.* **4**, 906–913 (2021).
Yang, W. *et al.* The Integration of Sub-10 nm Gate Oxide on MoS₂ with Ultra Low Leakage
and Enhanced Mobility. *Sci. Rep.* **5**, 11921 (2015).
Lee, G.-H. *et al.* Highly Stable, Dual-Gated MoS₂ Transistors Encapsulated by Hexagonal
Boron Nitride with Gate-Controllable Contact, Resistance, and Threshold Voltage. *ACS Nano*
**9**, 7019–7026 (2015).
Huang, J. K. *et al.* High- κ perovskite membranes as insulators for two-dimensional transistors.
*Nature* **605**, 262–267 (2022).
Illarionov, Y. Y. *et al.* Ultrathin calcium fluoride insulators for two-dimensional field-effect
transistors. *Nat. Electron.* **2**, 230–235 (2019).
Jiang, T. *et al.* Valley and band structure engineering of folded MoS₂ bilayers. *Nat.*
*Nanotechnol.* **9**, 825–829 (2014).
Liang, J. *et al.* Monitoring Local Strain Vector in Atomic-Layered MoSe₂ by Second-
Harmonic Generation. *Nano Lett.* **17**, 7539–7543 (2017).
Zhou, L. *et al.* Unconventional excitonic states with phonon sidebands in layered silicon
diphosphide. *Nat. Mater.* **21**, 773–778 (2022).
Dai, X. *et al.* Selective substitution induced anomalous phonon stiffening within quasi-one-
dimensional P—P chains in SiP₂. *Nano Research* **16**, 1107–1114 (2022).

REVIEWERS' COMMENTS

Reviewer #1 (Remarks to the Author):

the authors have carefully considered my questions and revised the manuscript properly. I support the publication of the revised paper.

Reviewer #2 (Remarks to the Author):

I am fully satisfied with the authors response and final version, and I recommend publication.

Reviewer #3 (Remarks to the Author):

The Authors have addressed all the comments in the revised version and improved the overall quality of the manuscript accordingly. The morphological studies via PFM or KPFM could be a future task; the current twist-angle-dependent PL data serves as a piece of important evidence to prove the origin of the anisotropic optical properties, i.e., moiré lattice.

Given the originality and quality of this work presented in this revised version, the manuscript should make a good contribution to progressing symmetry-breaking studies of 2D materials and thus should be a good fit in Nature Communications.

**Reviewer #1 (Remarks to the Author)**

*The authors have carefully considered my questions and revised the manuscript properly. I support the*
*publication of the revised paper.*

**Authors' response:**

We thank the reviewer for his/her careful review and constructive suggestion of our paper. We are glad
that our revised manuscript have satisfied him/her and appreciate his/her positive comments and
recommendation for publication of this work in *Nature Communications*.

**Reviewer #2 (Remarks to the Author)**

*I am fully satisfied with the authors response and final version, and I recommend publication.*

**Authors' response:**

We thank the reviewer for his/her careful review and constructive suggestion of our paper. We are glad
that our revised manuscript have satisfied him/her and appreciate his/her positive comments and
recommendation for publication of this work in *Nature Communications*.

**Reviewer #3 (Remarks to the Author)**

*The Authors have addressed all the comments in the revised version and improved the overall quality*
*of the manuscript accordingly. The morphological studies via PFM or KPFM could be a future task;*
*the current twist-angle-dependent PL data serves as a piece of important evidence to prove the origin*
*of the anisotropic optical properties, i.e., moiré lattice.*

*Given the originality and quality of this work presented in this revised version, the manuscript should*
*make a good contribution to progressing symmetry-breaking studies of 2D materials and thus should*
*be a good fit in Nature Communications.*

**Authors' response:**

We thank the reviewer for his/her careful review and constructive suggestion of our paper. We are glad
that our revised manuscript have satisfied him/her and appreciate his/her positive comments and
recommendation for publication of this work in *Nature Communications*.